# Stochastic Gradient Push for Distributed Deep Learning

## Abstract

Large mini-batch parallel SGD is commonly used for distributed training of deep networks. Approaches that use tightly-coupled exact distributed averaging based on AllReduce are sensitive to slow nodes and high-latency communication. In this work we show the applicability of Stochastic Gradient Push (SGP) for distributed training. SGP uses a gossip algorithm called PushSum for approximate distributed averaging, allowing for much more loosely coupled communications which can be beneficial in high-latency or high-variability scenarios. The tradeoff is that approximate distributed averaging injects additional noise in the gradient which can affect the train and test accuracies. We prove that SGP converges to a stationary point of smooth, non-convex objective functions. Furthermore, we validate empirically the potential of SGP. For example, using 32 nodes with 8 GPUs per node to train ResNet-50 on ImageNet, where nodes communicate over 10Gbps Ethernet, SGP completes 90 epochs in around 1.6 hours while AllReduce SGD takes over 5 hours, and the top-1 validation accuracy of SGP remains within 1.2% of that obtained using AllReduce SGD.

## 1 Introduction

Deep Neural Networks (DNNs) are the state-of-the art machine learning approach in many application areas, including image recognition (He et al., 2016) and natural language processing (Vaswani et al., 2017). Stochastic Gradient Descent (SGD) is the current workhorse for training neural networks. The algorithm optimizes the network parameters, $x$, to minimize a loss function, $f(\cdot)$, through gradient descent, where the loss function's gradients are approximated using a subset of training examples (a mini-batch). DNNs often require large amounts of training data and trainable parameters, necessitating non-trivial computational requirements (Wu et al., 2016; Mahajan et al., 2018). There is a need for efficient methods to train DNNs in large-scale computing environments.

A parallel version of SGD is usually adopted for large-scale, distributed training (Goyal et al., 2017; Li et al., 2014). Worker nodes compute local mini-batch gradients of the loss function on different subsets of the data, and then calculate an exact inter-node average gradient using either the ALLRE-DUCE communication primitive, in synchronous implementations (Goyal et al., 2017), or using a central parameter server, in asynchronous implementations (Dean et al., 2012). Using a parameter server to aggregate gradients introduces a potential bottleneck and a central point of failure (Lian et al., 2017). The ALLREDUCE primitive computes the exact average gradient at all workers in a decentralized manner, avoiding issues associated with centralized communication and computation.

However, exact averaging algorithms like ALLREDUCE are not robust in high-latency or high-variability platforms, *e.g.*, where the network bandwidth may be a significant bottleneck, because they involve tightly-coupled, blocking communication (*i.e.*, the call does not return until all nodes have finished aggregating). Moreover, aggregating gradients across all the nodes in the network can introduce non-trivial computational overhead when there are many nodes, or when the gradients themselves are large. This issue motivates the investigation of a decentralized and inexact version of SGD to reduce the overhead associated with distributed training.

There have been numerous decentralized optimization algorithms proposed and studied in the control-systems literature that leverage consensus-based approaches to aggregate information; see the recent survey Nedić et al. (2018) and references therein. Rather than exactly aggregating gradi-

ents (as with ALLREDUCE), this line of work uses less-coupled message passing algorithms which compute inexact distributed averages.

Most previous work in this area has focused on theoretical convergence analysis assuming convex objectives. Recent work has begun to investigate their applicability to large-scale training of DNNs (Lian et al., 2017; Jiang et al., 2017). However, these papers study methods based on communication patterns which are static (the same at every iteration) and symmetric (if $i$ sends to $j$, then $i$ must also receive from $j$ before proceeding). Such methods inherently require blocking and communication overhead. State-of-the-art consensus optimization methods build on the PUSHSUM algorithm for approximate distributed averaging (Kempe et al., 2003; Nedić et al., 2018), which allows for non-blocking, time-varying, and directed (asymmetric) communication. Since SGD already uses stochastic mini-batches, the hope is that an inexact average mini-batch will be as useful as the exact one if the averaging error is sufficiently small relative to the variability in the stochastic gradient.

This paper studies the use of Stochastic Gradient Push (SGP), an algorithm blending SGD and PUSHSUM, for distributed training of deep neural networks. We provide a theoretical analysis of SGP, showing it converges for smooth non-convex objectives. We also evaluate SGP experimentally, training ResNets on ImageNet using up to 32 nodes, each with 8 GPUs (*i.e.*, 256 GPUs in total). Our main contributions are summarized as follows:

- We provide the first convergence analysis for Stochastic Gradient Push when the objective function is smooth and non-convex. We show that, for an appropriate choice of the step size, SGP converges to a stationary point at a rate of $\mathcal{O}\left(1/\sqrt{nK}\right)$, where $n$ is the number of nodes and $K$ is the number of iterations.

- In a high-latency scenario, where nodes communicate over 10Gbps Ethernet, SGP runs up to $3\times$ faster than ALLREDUCE SGD and exhibits 88.6% scaling efficiency over the range from 4–32 nodes.

- The top-1 validation accuracy of SGP matches that of ALLREDUCE SGD for up to 8 nodes (64 GPUs), and remains within 1.2% of ALLREDUCE SGD for larger networks.

- In a low-latency scenario, where nodes communicate over a 100Gbps InfiniBand network supporting GPUDirect, SGP is on par with ALLREDUCE SGD in terms of running time, and SGP exhibits 92.4% scaling efficiency.

- In comparison to other synchronous decentralized consensus-based approaches that require symmetric messaging, SGP runs faster and it produces models with better validation accuracy.

## 2 PRELIMINARIES

**Problem formulation.** We consider the setting where a network of $n$ nodes cooperates to solve the stochastic consensus optimization problem

$$\begin{aligned} \min_{\boldsymbol{x}_i \in \mathbb{R}^d, i=1,\ldots,n} \quad & \frac{1}{n}\sum_{i=1}^n \mathbb{E}_{\xi_i \sim D_i} F_i(\boldsymbol{x}_i; \xi_i) \\ \text{subject to} \quad & \boldsymbol{x}_i = \boldsymbol{x}_j, \forall i,j = 1,\ldots,n. \end{aligned} \quad (1)$$

Each node has local data following a distribution $D_i$, and the nodes wish to cooperate to find the parameters $\boldsymbol{x}$ of a DNN that minimizes the average loss with respect to their data, where $F_i$ is the loss function at node $i$. Moreover, the goal codified in the constraints is for the nodes to reach agreement (*i.e.*, consensus) on the solution they report. We assume that nodes can locally evaluate stochastic gradients $\nabla F(\boldsymbol{x}_i; \xi_i)$, $\xi_i \sim D_i$, but they must communicate to access information about the objective functions at other nodes.

**Distributed averaging.** The problem described above encompasses distributed training based on data parallelism. There a canonical approach is large mini-batch parallel stochastic gradient descent: for an overall mini-batch of size $nb$, each node computes a local stochastic mini-batch gradient using $b$ samples, and then the nodes use the ALLREDUCE communication primitive to compute the average gradient at every node. Let $f_i(\boldsymbol{x}_i) = \mathbb{E}_{\xi_i \sim D_i} F_i(\boldsymbol{x}_i; \xi_i)$ denote the objective at node $i$, and let $f(\boldsymbol{x}) = \frac{1}{n}\sum_{i=1}^n f_i(\boldsymbol{x})$ denote the overall objective. Since $\nabla f(\boldsymbol{x}) = \frac{1}{n}\sum_{i=1}^n \nabla f_i(\boldsymbol{x})$, averaging gradients via ALLREDUCE provides an exact stochastic gradient of $f$. Typical implementations of

ALLREDUCE have each node send and receive $2\frac{n-1}{n}$B bytes, where B is the size (in bytes) of the tensor being reduced, and involve $2\log_2(n)$ communication steps (Rabenseifner, 2004). Moreover, ALLREDUCE is a blocking primitive, meaning that no node will proceed with local computations until the primitive returns.

**Approximate distributed averaging.**  In this work we explore the alternative approach of using a gossip algorithm for approximate distributed averaging—specifically, the PUSHSUM algorithm. Gossip algorithms typically use linear iterations for averaging. For example, let $\boldsymbol{y}_i^{(0)} \in \mathbb{R}^n$ be a vector at node $i$, and consider the goal of computing the average vector $\frac{1}{n}\sum_{i=1}^n \boldsymbol{y}_i^{(0)}$ at all nodes. Stack the initial vectors into a matrix $\boldsymbol{Y}^{(0)} \in \mathbb{R}^{n\times d}$ with one row per node. Typical gossip iterations have the form $\boldsymbol{Y}^{(k+1)} = \boldsymbol{P}^{(k)}\boldsymbol{Y}^{(k)}$ where $\boldsymbol{P}^{(k)} \in \mathbb{R}^{n\times n}$ is referred to as the mixing matrix. This corresponds to the update $\boldsymbol{y}_i^{(k+1)} = \sum_{j=1}^n p_{i,j}^{(k)}\boldsymbol{y}_j^{(k)}$ at node $i$. To implement this update, node $i$ only needs to receive messages from other nodes $j$ for which $p_{i,j}^{(k)} \neq 0$, so it will be appealing to use sparse $\boldsymbol{P}^{(k)}$ to reduce communications.

Drawing inspiration from the theory of Markov chains (Seneta, 1981), the mixing matrices $\boldsymbol{P}^{(k)}$ are designed to be column stochastic. Then, under mild conditions (*e.g.*, ensuring that information from every node eventually reaches all other nodes) one can show that $\lim_{K\to\infty}\prod_{k=0}^K \boldsymbol{P}^{(k)} = \boldsymbol{\pi}\mathbf{1}^\top$, where $\boldsymbol{\pi}$ is the ergodic limit of the chain and $\mathbf{1}$ is a vector with all entries equal to 1. Consequently, the gossip iterations converge to a limit $\boldsymbol{Y}^{(\infty)} = \boldsymbol{\pi}\big(\mathbf{1}^\top\boldsymbol{Y}^{(0)}\big)$; *i.e.*, the value at node $i$ converges to $\boldsymbol{y}_i^{(\infty)} = \pi_i\sum_{j=1}^n \boldsymbol{y}_j^{(0)}$. When the matrices $\boldsymbol{P}^{(k)}$ are symmetric, it is straightforward to design the algorithm so that $\pi_i = 1/n$ for all $i$ by making $\boldsymbol{P}^{(k)}$ doubly stochastic. However, symmetric $\boldsymbol{P}^{(k)}$ has strong practical ramifications, such as requiring care in the implementation to avoid deadlocks.

The PUSHSUM algorithm only requires that $\boldsymbol{P}^{(k)}$ be column-stochastic, and not necessarily symmetric (so node $i$ may send to node $j$, but not necessarily vice versa). Instead, one additional scalar parameter $w_i^{(k)}$ is maintained at each node. The parameter is initialized to $w_i^{(0)} = 1$ for all $i$, and updated using the same linear iteration, $\boldsymbol{w}^{(k+1)} = \boldsymbol{P}^{(k)}\boldsymbol{w}^{(k)}$. Consequently, the parameter converges to $\boldsymbol{w}^{(\infty)} = \boldsymbol{\pi}(\mathbf{1}^\top\boldsymbol{w}^{(0)})$, or $w_i^{(\infty)} = \pi_i n$ at node $i$. Thus each node can recover the average of the initial vectors by computing the *de-biased* ratio $\boldsymbol{y}_i^{(\infty)}/w_i^{(\infty)}$. In practice, we stop after a finite number of gossip iterations $K$ and compute $\boldsymbol{y}_i^{(K)}/w_i^{(K)}$. The distance of the de-biased ratio to the exact average can be quantified in terms of properties of the matrices $\{\boldsymbol{P}^{(k)}\}_{k=0}^{K-1}$. Let $\mathcal{N}_i^{\text{out}(k)} = \{j\colon p_{j,i}^{(k)} > 0\}$ and $\mathcal{N}_i^{\text{in}(k)} = \{j\colon p_{i,j}^{(k)} > 0\}$ denote the sets of nodes that $i$ transmits to and receives from, respectively, at iteration $k$. If we use B bytes to represent the vector $\boldsymbol{y}_i^{(k)}$, then node $i$ sends and receives $\big|\mathcal{N}_i^{\text{out}(k)}\big|$B and $\big|\mathcal{N}_i^{\text{in}(k)}\big|$B bytes, respectively, per iteration. In our experiments we use graph sequences with $\big|\mathcal{N}_i^{\text{out}(k)}\big| = \big|\mathcal{N}_i^{\text{in}(k)}\big| = 1$ or 2, and find that approximate averaging is both fast and still facilitates training.

## 3  STOCHASTIC GRADIENT PUSH

**Algorithm description.**  The *stochastic gradient push* (SGP) method for solving equation 1 is obtained by interleaving one local stochastic gradient descent update at each node with one iteration of PUSHSUM. Each node maintains three variables: the model parameters $\boldsymbol{x}_i^{(k)}$ at node $i$, the scalar PUSHSUM weight $w_i^{(k)}$, and the de-biased parameters $\boldsymbol{z}_i^{(k)} = \big(w_i^{(k)}\big)^{-1}\boldsymbol{x}_i^{(k)}$. The initial $\boldsymbol{x}_i^{(0)}$ and $\boldsymbol{z}_i^{(0)}$ can be initialized to any arbitrary value as long as $\boldsymbol{x}_i^{(0)} = \boldsymbol{z}_i^{(0)}$. Pseudocode is shown in Alg. 1. Each node performs a local SGD step (lines 2–4) followed by one step of PUSHSUM for approximate distributed averaging (lines 5–8).

Note that the gradients are evaluated at the de-biased parameters $\boldsymbol{z}_i^{(k)}$ in line 3, and they are then used to update $\boldsymbol{x}_i^{(k)}$, the PUSHSUM numerator, in line 4. All communication takes place in line 5, and each message contains two parts, the PUSHSUM numerator and denominator. In particular, node $i$ controls the values $p_{j,i}^{(k)}$ used to weight the values in messages it sends.

---

**Algorithm 1** Stochastic Gradient Push (SGP)

---

**Require:** Initialize $\gamma > 0$, $\boldsymbol{x}_i^{(0)} = \boldsymbol{z}_i^{(0)} \in \mathbb{R}^d$ and $w_i^{(0)} = 1$ for all nodes $i \in \{1, 2, \ldots, n\}$

1: **for** $k = 0, 1, 2, \cdots, K$ **do** at node $i$
2:     Sample new mini-batch $\xi_i^{(k)} \sim \mathcal{D}_i$ from local distribution
3:     Compute a local stochastic mini-batch gradient at $\boldsymbol{z}_i^{(k)}$: $\nabla \boldsymbol{F}_i(\boldsymbol{z}_i^{(k)}; \xi_i^{(k)})$
4:     $\boldsymbol{x}_i^{(k+\frac{1}{2})} = \boldsymbol{x}_i^{(k)} - \gamma \nabla \boldsymbol{F}_i(\boldsymbol{z}_i^{(k)}; \xi_i^{(k)})$
5:     Send $\left(p_{j,i}^{(k)} \boldsymbol{x}_i^{(k+\frac{1}{2})}, p_{j,i}^{(k)} w_i^{(k)}\right)$ to out-neighbors $j \in \mathcal{N}_i^{\text{out}(k)}$;
       receive $\left(p_{i,j}^{(k)} \boldsymbol{x}_j^{(k+\frac{1}{2})}, p_{i,j}^{(k)} w_j^{(k)}\right)$ from in-neighbors $j \in \mathcal{N}_i^{\text{in}(k)}$
6:     $\boldsymbol{x}_i^{(k+1)} = \sum_{j \in \mathcal{N}_i^{\text{in}(k)}} p_{i,j}^{(k)} \boldsymbol{x}_j^{(k+\frac{1}{2})}$
7:     $w_i^{(k+1)} = \sum_{j \in \mathcal{N}_i^{\text{in}(k)}} p_{i,j}^{(k)} w_j^{(k)}$
8:     $\boldsymbol{z}_i^{(k+1)} = \boldsymbol{x}_i^{(k+1)} / w_i^{(k+1)}$
9: **end for**

---

We are mainly interested in the case where the mixing matrices $\boldsymbol{P}^{(k)}$ are sparse in order to have low communication overhead. However, we point out that when the nodes' initial values are identical, $\boldsymbol{x}_i^{(0)} = \boldsymbol{x}_j^{(0)}$ for all $i, j \in [n]$, and every entry of $\boldsymbol{P}^{(k)}$ is equal to $1/n$, then SGP is mathematically equivalent to parallel SGD using ALLREDUCE. Please refer to appendix A for pratical implementation details, including how we design mixing matrices $\boldsymbol{P}^{(k)}$.

**Theoretical guarantees.** SGP was first proposed and analyzed in (Nedić & Olshevsky, 2016) assuming the local objectives $f_i(\boldsymbol{x})$ are strongly convex. Here we provide convergence results in the more general setting of smooth, non-convex objectives. We make the following three assumptions:

1. ($L$-smooth) There exists a constant $L > 0$ such that $\|\nabla f_i(\boldsymbol{x}) - \nabla f_i(\boldsymbol{y})\| \le L\|\boldsymbol{x} - \boldsymbol{y}\|$, or equivalently

$$f_i(\boldsymbol{x}) \le f_i(\boldsymbol{y}) + \nabla f_i(\boldsymbol{y})^\top (\boldsymbol{x} - \boldsymbol{y}) + \frac{L}{2}\|\boldsymbol{y} - \boldsymbol{x}\|^2. \tag{2}$$

Note that this assumption implies that function $f(x)$ is also L-smooth.

2. (Bounded variance) There exist finite positive constants $\sigma^2$ and $\zeta^2$ such that

$$\mathbb{E}_{\xi \sim D_i} \|\nabla F_i(\boldsymbol{x}; \xi) - \nabla f_i(\boldsymbol{x})\|^2 \le \sigma^2 \quad \forall i, \forall \boldsymbol{x}, \text{ and} \tag{3}$$

$$\frac{1}{n} \sum_{i=1}^n \|\nabla f_i(\boldsymbol{x}) - \nabla f(\boldsymbol{x})\|^2 \le \zeta^2 \quad \forall \boldsymbol{x}. \tag{4}$$

Thus $\sigma^2$ bounds the variance of stochastic gradients at each node, and $\zeta^2$ quantifies the similarity of data distributions at different nodes.

3. (Mixing connectivity) To each mixing matrix $\boldsymbol{P}^{(k)}$ we can associate a graph with vertex set $\{1, \ldots, n\}$ and edge set $E^{(k)} = \{(i,j) : p_{i,j}^{(k)} > 0\}$; *i.e.*, with edges $(i,j)$ from $j$ to $i$ if $i$ receives a message from $j$ at iteration $k$. Assume that the graph with edge set $\bigcup_{k=lB}^{(l+1)B-1} E^{(k)}$ is strongly connected and has diameter at most $\Delta$ for every $l \ge 0$. To simplify the discussion, we assume that every column of the mixing matrices $\boldsymbol{P}^{(k)}$ has at most $D$ non-zero entries.

Let $\overline{\boldsymbol{x}}^{(k)} = \frac{1}{n} \sum_{i=1}^n \boldsymbol{x}_i^{(k)}$. Under similar assumptions, Lian et al. (2017) define that a decentralized algorithm for solving equation 1 converges if, for any $\epsilon > 0$, it eventually satisfies

$$\frac{1}{K} \sum_{k=1}^K \mathbb{E}\|\nabla f(\overline{\boldsymbol{x}}^{(k)})\|^2 \le \epsilon. \tag{5}$$

Our first result shows that SGP converges in this sense.

**Theorem 1.** *Suppose that Assumptions 1–3 hold, and run SGP for $K$ iterations with step-size $\gamma = \sqrt{n/K}$. Let $f^* = \min_{\boldsymbol{x}} f(\boldsymbol{x})$ and assume that $f^* > -\infty$. There exist constants $C > 0$ and*

$q \in (0, 1)$ *which depend on* $B$, $n$, *and* $\Delta$ *such that if the total number of iterations satisfies*

$$K \geq \max \left\{ n, \frac{nL^4C^460^2}{(1-q)^4}, \frac{L^4C^4P_1^2n}{(1-q)^4(f(\overline{\boldsymbol{x}}^{(0)}) - f^* + \frac{L\sigma^2}{2})^2}, \frac{L^2C^2nP_2}{(1-q)^2(f(\overline{\boldsymbol{x}}^{(0)}) - f^* + \frac{L\sigma^2}{2})} \right\} \tag{6}$$

*where* $P_1 = 4(\sigma^2 + 3\zeta^2)n + \frac{\sum_{i=1}^n \left\|\boldsymbol{x}_i^{(0)}\right\|^2}{n}$ *and* $P_2 = \sigma^2 + 3\zeta^2L^2C^2 + 2\frac{\sum_{i=1}^n \left\|\boldsymbol{x}_i^{(0)}\right\|^2}{n}$, *then*

$$\frac{\sum_{k=0}^{K-1} \mathbb{E} \left\|\nabla f(\overline{\boldsymbol{x}}^{(k)})\right\|^2}{K} \leq \frac{12(f(\overline{\boldsymbol{x}}^{(0)}) - f^* + \frac{L\sigma^2}{2})}{\sqrt{nK}}.$$

The proof is given in Appendix C, where we also provide precise expressions for the constants $C$ and $q$. The proof of Theorem 1 builds on an approach developed in Lian et al. (2017). Theorem 1 shows that, for a given number of nodes $n$, by running a sufficiently large number of iterations $K$ (roughly speaking, $\Omega(n)$, which is reasonable for distributed training of DNNs) and choosing the step-size $\gamma$ as prescribed, then the criterion equation 5 is satisfied with a number of iterations $K = \Omega(1/n\epsilon^2)$. That is, we achieve a linear speedup in the number of nodes.

Theorem 1 shows that the average of the nodes parameters, $\overline{\boldsymbol{x}}^{(k)}$, converges, but it doesn't directly say anything about the parameters at each node. In fact, we can show a stronger result.

**Theorem 2.** *Under the same assumptions as in Theorem 1,*

$$\frac{1}{nK} \sum_{k=0}^{K-1} \sum_{i=1}^n \mathbb{E} \left\|\overline{\boldsymbol{x}}^{(k)} - \boldsymbol{z}_i^{(k)}\right\|^2 \leq \mathcal{O}\left(\frac{1}{K} + \frac{1}{K^{3/2}}\right).$$

*and*

$$\frac{1}{nK} \sum_{k=0}^{K-1} \sum_{i=1}^n \mathbb{E} \left\|\nabla f(\boldsymbol{z}_i^k)\right\|^2 \leq O\left(\frac{1}{\sqrt{nK}} + \frac{1}{K} + \frac{1}{K^{3/2}}\right)$$

The proof is also given in Appendix C. This result shows that as $K$ grows, the de-biased variables $\boldsymbol{z}_i^{(k)}$ converge to the node-wise average $\overline{\boldsymbol{x}}^{(k)}$, and hence the de-biased variables at each node also converge to a stationary point. Note that for fixed $n$ and large $K$, the $1/\sqrt{nK}$ term will dominate the other factors.

## 4 RELATED WORK

A variety of approaches have been proposed to accelerate distributed training of DNNs, including quantizing gradients (Alistarh et al., 2007; Wen et al., 2007) and performing multiple local SGD steps at each node before averaging (McMahan et al., 2017). These approaches are complementary to the tradeoff we consider in this paper, between exact and approximate distributed averaging. Similar to using PUSHSUM for averaging, both quantizing gradients and performing multiple local SGD steps before averaging can also be seen as injecting additional noise into SGD, leading to a trade off between training faster (by reducing communication overhead) and potentially obtaining a less accurate result. Combining these approaches (quantized, inexact, and infrequent averaging) is an interesting direction for future work.

For the remainder of this section we review related work applying consensus-based approaches to large-scale training of DNNs. Blot et al. (2016) report initial experimental results on small-scale experiments with an SGP-like algorithm. Jin et al. (2016) make a theoretical connection between PUSHSUM-based methods and Elastic Averaging SGD (Zhang et al., 2015). Relative to those previous works, we provide the first convergence analysis for a PUSHSUM-based method in the smooth non-convex case. Lian et al. (2017) and Jiang et al. (2017) study synchronous consensus-based versions of SGD. However, unlike PUSHSUM, those methods involve symmetric message passing (if $i$ sends to $j$ at iteration $k$, then $j$ also sends to $i$ before both nodes update) which is inherently blocking. Consequently, these methods are more sensitive to high-latency communication settings, and each node generally must communicate more per iteration, in comparison to PUSHSUM-based SGP where communication may be directed ($i$ can send to $j$ without needing a response from $i$). The decentralized parallel SGD (D-PSGD) method proposed in Lian et al. (2017) produces iterates whose

node-wise average, $\overline{x}^{(k)}$, is shown to converge in the sense of equation 5. Our proof of Theorem 1, showing the convergence of SGP in the same sense, adapts some ideas from their analysis and also goes beyond to show that, since the values at each node converge to the average, the individual values at each node also converge to a stationary point. We compare SGP with D-PSGD experimentally in Section 5 below and find that although the two methods find solutions of comparable accuracy, SGP is consistently faster.

Jin et al. (2016) and Lian et al. (2018) study asynchronous consensus-based methods for training DNNs. Lian et al. (2018) analyzes an asynchronous version of D-PSGD and proves that its node-wise averages also converge to a stationary point. In general, these contributions focusing on asynchrony can be seen as orthogonal to the use of a PUSHSUM based protocol for consensus averaging.

## 5 EXPERIMENTS

Next, we compare SGP with ALLREDUCE SGD, and D-PSGD (Lian et al., 2017), an approximate distributed averaging baseline relying on doubly-stochastic gossip. We run experiments on a large-scale distributed computing environment using up to 256 GPUs. Our results show that when communication is the bottleneck, SGP is faster than both SGD and D-PSGD. SGP also outperforms D-PSGD in terms of validation accuracy, while achieving a slightly worse accuracy compared to SGD when using a large number of compute nodes. Our results also highlight that, in a setting where communication is efficient (*e.g.*, over InfiniBand), doing exact averaging through ALLREDUCE SGD remains a competitive approach.

We run experiments on 32 DGX-1 GPU servers in a high-performance computing cluster. Each server contains 8 NVIDIA Volta-V100 GPUs. We consider two communication scenarios: in the *high-latency* scenario the nodes communicate over a 10 Gbit/s Ethernet network, and in the *low-latency* scenario the nodes communicate over 100 Gbit/s InfiniBand, which supports GPUDirect RDMA communications. To investigate how each algorithm scales, we run experiments with 4, 8, 16, and 32 nodes (*i.e.*, 32, 64, 128, and 256 GPUs).

We adopt the 1000-way ImageNet classification task (Russakovsky et al., 2015) as our experimental benchmark. We train a ResNet-50 (He et al., 2016) following the experimental protocol of Goyal et al. (2017), using the same hyperparameters with the exception of the learning rate schedule in the 32 node experiment for SGP and D-PSGD. In the experiments, we also modify SGP to use Nesterov momentum. In our default implementation of SGP, each node sends and receives to one other node at each iteration, and this destination changes from one iteration to the next. Please refer to appendix A for more information about our implementation, including how we design/implement the sequence of mixing matrices $\boldsymbol{P}^{(k)}$.

All algorithms are implemented in PyTorch v0.5 (Paszke et al.). To leverage the highly efficient NVLink interconnect within each server, we treat each DGX-1 as one node in all of our experiments. In our implementation of SGP, each node computes a local mini-batch in parallel using all eight GPUs using a local ALLREDUCE, which is efficiently implemented via the NVIDIA Collective Communications Library. Then inter-node averaging is accomplished using PUSHSUM either over Ethernet or InfiniBand. In the low-latency experiments, we leverage GPUDirect to directly send/receive messages between GPUs on different nodes and avoid transferring the model back to host memory. In the high-latency experiments this is not possible, so the model is transferred to host memory after the local ALLREDUCE, and then PUSHSUM messages are sent over Ethernet.

### 5.1 EVALUATION ON HIGH-LATENCY INTERCONNECT

We consider the high-latency scenario where nodes communicate over 10Gbit/s Ethernet. With a local mini-batch size of 256 samples per node (32 samples per GPU), a single Volta DGX-1 server can perform roughly $4.384$ mini-batches per second. Since the ResNet-50 model size is roughly 100MBytes, transmitting one copy of the model per iteration requires 3.5 Gbit/s. Thus in the high-latency scenario the problem, if a single 10 Gbit/s link must carry the traffic between more than two pairs of nodes, then communication clearly becomes a bottleneck.

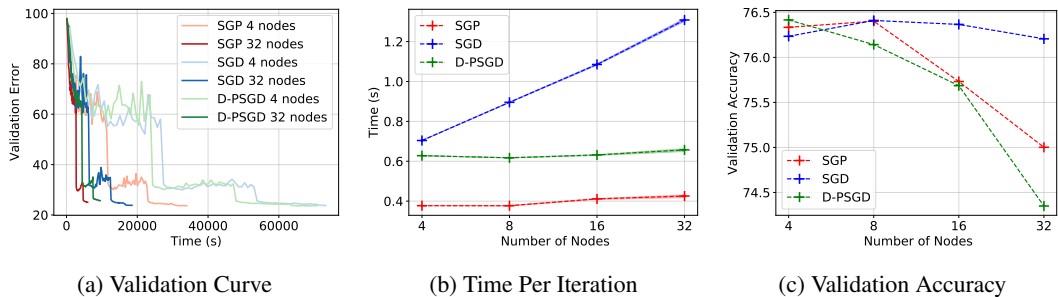

(a) Validation Curve      (b) Time Per Iteration      (c) Validation Accuracy

Figure 1: Results on Ethernet 10Gbits. (a): Validation performance w.r.t. training time (in seconds) for model trained on 4 and 32 nodes. (b): Average time per training iteration (in seconds) (c): Best validation accuracy. Stochastic Gradient Push (SGP) is faster than both Decentralized-Parallel SGD (D-PSGD) and ALLREDUCE SGD while decreasing validation accuracy by 1.2%.

**Comparison with synchronous approaches.**  We first compare SGP with other synchronous and decentralized approaches. Figure 1 (a) shows the validation curves when training on 4 and 32 nodes (additional training and validation curves for all the training runs can be found in B.1). Note that when we increase the number of nodes $n$, we also decrease the total number of iterations $K$ to $K/n$ following Theorem 1 (see Figure B.3). For any number of nodes used in our experiments, we observe that SGP consistently outperforms D-PSGD and ALLREDUCE SGD in terms of total training time in this scenario. In particular for 32 nodes, SGP training time takes less than 1.6 hours while D-PSGD and ALLREDUCE SGD require roughly 2.6 and 5.1 hours. Appendix B.2 provides experimental evidence that all nodes converge to models with a similar training and validation accuracy when using SGP.

Figure 1 (b) shows the average time per iteration for the different training runs. As we increase the number of nodes, the average iteration time stays almost constant for SGP and D-PSGD, while we observe a significant time-increase in the case of ALLREDUCE SGD, resulting in an overall slower training time. Moreover, although D-PSGD and SGP both exhibit strong scaling, SGP is roughly 200ms faster per iteration, supporting the claim that it involves less communication overhead.

Figure 1 (c) reports the best validation accuracy for the different training runs. While they all start around the same value, the accuracy of D-PSGD and SGP decreases as we increase the number of nodes. In the case of SGP, we see its performance decrease by 1.2% relative to SGD on 32 nodes. We hypothesize that this decrease is due to the noise introduced by approximate distributed averaging. We will see below than changing the connectivity between the nodes can ameliorate this issue. We also note that the SGP validation accuracy is better than D-PSGD for larger networks.

**Comparison with asynchronous approach.**  The results in Tables 1 and 2 provide a comparison between the aforementioned synchronous methods and AD-PSGD (Lian et al., 2018), a state-of-art asynchronous method. AD-PSGD is an asynchronous implementation of the doubly-stochastic method D-PSGD, which relies on doubly-stochastic averaging. All methods are trained for exactly 90 epochs, therefore, the time-per-iteration is a direct reflection of the total training time. Training using AD-PSGD does not degrade the accuracy (relative to D-PSGD), and provides substantial speedups in training time. Relative to SGP, the AD-PSGD method runs slightly faster at the expense of lower validation accuracy (except in the 32 nodes case). In general, we emphasize that this asynchronous line of work is orthogonal, and that by combining the two approaches (leveraging the PUSHSUM protocol in an asynchronous manner), one can expect to further speed up SGP. We leave this as a promising line of investigation for future work.

## 5.2 EVALUATION ON A "LOW LATENCY" INTERCONNECT

We now investigate the behavior of SGP and ALLREDUCE SGD over InfiniBand 100Gbit/s, following the same experimental protocol as in the Ethernet 10Gbit/s case. In this scenario which is not

|                 | 4 nodes | 8 nodes | 16 nodes | 32 nodes |
|-----------------|---------|---------|----------|----------|
| ALLREDUCE SGD   | 76.23   | 76.41   | 76.37    | 76.21    |
| D-PSGD          | 76.42   | 76.14   | 75.69    | 74.35    |
| AD-PSGD         | 76.07   | 75.96   | 75.51    | 74.98    |
| SGP             | 76.33   | 76.40   | 75.73    | 75.00    |

Table 1: Top-1 Validation accuracy (%) over 10Gbps Ethernet showcasing an additional comparison with the AD-PSGD asynchronous doubly-stochastic approach.

|                 | 4 nodes | 8 nodes | 16 nodes | 32 nodes |
|-----------------|---------|---------|----------|----------|
| ALLREDUCE SGD   | 0.704   | 0.896   | 1.086    | 1.308    |
| D-PSGD          | 0.628   | 0.618   | 0.632    | 0.657    |
| AD-PSGD         | 0.361   | 0.363   | 0.374    | 0.388    |
| SGP             | 0.377   | 0.377   | 0.411    | 0.426    |

Table 2: Average time per iteration (seconds) over 10Gbps Ethernet showcasing an additional comparison with the AD-PSGD asynchronous doubly-stochastic approach. The average time per iteration for the asynchronous method is calculated by dividing the average time per epoch by the total number of iterations per epoch.

communication bound for a Resnet-50 model, we do not expect SGP to outperform ALLREDUCE SGD. Our goal is to illustrate that SGP is not significantly slower than ALLREDUCE SGD.

On this low-latency interconnect, SGD and SGP obtain similar timing and differ at most by 21ms per iteration (Figure 2 (b) for 4 nodes). In particular, using 32 nodes, SGP trains a ResNet-50 on ImageNet in 1.16 hours and SGD in 1.20 hours. SGD, however, exhibits better validation accuracy for large networks. Communication on InfiniBand is not a bottleneck for models the size of ResNet-50. These results therefore confirm that SGP benefits are more prominent in high-latency/low-bandwidth communication-bound scenarios. Although timing are similar, SGP still shows better scaling in term of sample throughput than ALLREDUCE SGD (see Figure B.6)

For experiments running at this speed (less than 0.31 seconds per iteration), timing could be impacted by other factors such as data loading. To better isolate the effects of data-loading, we run additional experiments on 32, 64, and 128 GPUs where we first copied the data locally on every node; see Appendix B.3 for more details. In that setting, the time-per-iteration of SGP remains approximately constant as we increase the number of nodes in the network, while the time for AllReduceSGD increases with more nodes.

## 5.3 IMPACT OF GRAPH TOPOLOGY

Next we investigate the impact of the communication graph topology on the SGP validation performance using Ethernet 10Gbit/s. In the limit of a fully-connected communication graph, SGD and SGP are strictly equivalent (see section 3). By increasing the number of neighbors in the graph, we expect the accuracy of SGP to improve (approximate averages are more accurate) but the communication time required for training will increase.

In Figure 3, we compare the training and validation accuracies of SGP using a communication graph with 1-neighbor and 2-neighbors with D-PSGD and SGD on 32 nodes. By increasing the number of neighbors to two, SGP achieves better training/validation accuracy (from 74.8/75.0 to 75.6/75.4) and gets closer to final validation achieves by SGD (77.0/76.2). Increasing the number of neighbors also increases the communication, hence the overall training time. SGP with 2 neighbors completes training in 2.1 hours and its average time per iteration increases by 27% relative to SGP with one neighbor. Nevertheless, SGP 2-neighbors is still faster than SGD and D-PSGD, while achieving better accuracy than SGP 1-neighbor.

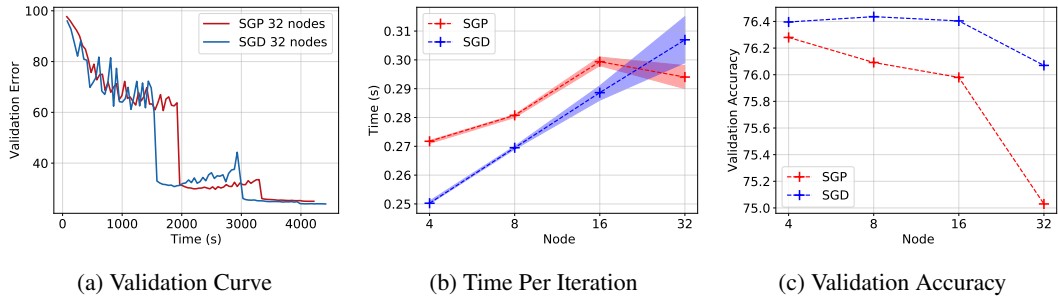

(a) Validation Curve     (b) Time Per Iteration     (c) Validation Accuracy

Figure 2: Results on InfiniBand 100Gbits. (a): Validation performance w.r.t. training time (in second) for model trained on 32 nodes. (b): Average time per training iteration (in second) (c): Best validation accuracy. Stochastic Gradient Push (SGP) is on par and sometime even slightly faster than ALLREDUCE SGD on "low latency" network while slightly degrading the accuracy.

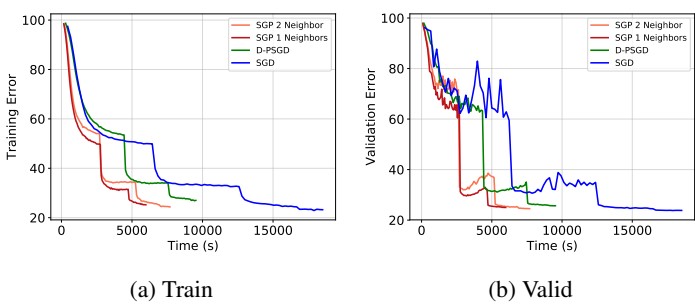

(a) Train       (b) Valid

Figure 3: Comparison of SGP using a communication graph with 1-neighbor, SGP using a graph with 2-neighbors, D-PSGD and SGD on 32 nodes communicating over 10 Gbit/s Ethernet. Using one additional neighbor improves the validation performance of SGD (from 75.0 to 75.4) while retaining most of the computational benefits.

## 6 CONCLUSION

DNN training often necessistates non-trivial computational requirements leveaging distributed computing resources. Traditional parallel versions of SGD use exact averaging algorithms to parallelize the computation between nodes, and induce additional parallelization overhead as the model and network sizes grow. This paper proposes the use of Stochastic Gradient Push for distributed deep learning. The proposed method computes in-exact averages at each iteartion in order to improve scaling efficiency and reduce the dependency on the underlying network topology. SGP converges to a stationary point at an $\mathcal{O}\left(1/\sqrt{nK}\right)$ rate in the smooth and non-convex case, and proveably achieves a linear speedup (in iterations) with respect to the number of nodes. Empirical results show that SGP can be up to $3\times$ times faster than traditional ALLREDUCE SGD over high-latency interconnect, matches the top-1 validation accuracy up to 8 nodes (64GPUs), and remains within $1.2\%$ of the top-1 validation accuracy for larger-networks.

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

# A    IMPLEMENTATION DETAILS

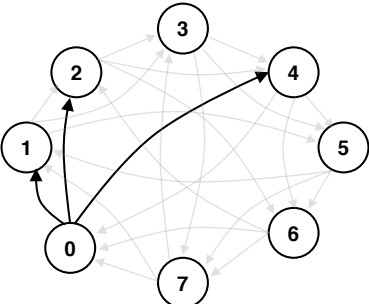

(a) Directed Exponential Graph high-
lighting node 0's out-neighbours

Figure A.1: Example of an 8-node exponential graph used in experiments

## A.1    COMMUNICATION TOPOLOGY

**Directed exponential graph.**    For the SGP experiments we use a time-varying directed graph to represent the inter-node connectivity. Thinking of the nodes as being ordered sequentially, according to their rank, $0, \ldots, n-1$,[1] each node periodically communicates with peers that are $2^0, 2^1, \ldots, 2^{\lfloor \log_2(n-1) \rfloor}$ hops away. Fig. A.1 shows an example of a directed 8-node exponential graph. Node 0's $2^0$-hop neighbour is node 1, node 0's $2^1$-hop neighbour is node 2, and node 0's $2^2$-hop neighbour is node 4.

In the one-peer-per-node experiments, each node cycles through these peers, transmitting, only, to a single peer from this list at each iteration. E.g., at iteration $k$, all nodes transmit messages to their $2^0$-hop neighbours, at iteration $k+1$ all nodes transmit messages to their $2^1$-hop neighbours, an so on, eventually returning to the beginning of the list before cycling through the peers again. This procedure ensures that each node only sends and receives a single message at each iteration. By using full-duplex communication, sending and receiving can happen in parallel.

In the two-peer-per-node experiments, each node cycles through the same set of peers, transmitting to two peers from the list at each iteration. E.g., at iteration $k$, all nodes transmit messages to their $2^0$-hop and $2^1$-hop neighbours, at iteration $k+1$ all nodes transmit messages to their $2^1$-hop and $2^2$ neighbours, an so on, eventually returning to the beginning of the list before cycling through the peers again. Similarly, at each iteration, each node also receives, in a full-duplex manner, two messages from some peers that are unknown to the receiving node ahead of time. Thereby performing the send and receive operations in parallel.

**Definition of $\boldsymbol{P}^{(k)}$.**    Based on the description above, in the one-peer-per-node experiments, each node sends to one neighbor at every iteration, and so each column of $\boldsymbol{P}^{(k)}$ has exactly two non-zero entries, both of which are equal to $1/2$. The diagonal entries $p_{i,i}^{(k)} = 1/2$ for all $i$ and $k$. At time step $k$, each node sends to a neighbor that is $2^{k \bmod \lfloor \log_2(n-1) \rfloor}$ hops away. Thus, with $h_k = 2^{k \bmod \lfloor \log_2(n-1) \rfloor}$, we get that

$$p_{j,i}^{(k)} = \begin{cases} 1/2, & \text{if } j = (i + h_k) \bmod n \\ 0, & \text{otherwise.} \end{cases}$$

Note that, with this design, in fact each node sends to one peer and receives from one peer at every iteration, so the communication load is balanced across the network.

In the two-peer-per-node experiments, the definition is similar, but now there will be three non-zero entries in each column of $\boldsymbol{P}^{(k)}$, all of which will be equal to $1/3$; these are the diagonal, and the entries corresponding to the two neighbors to which the node sends at that iteration. In addition, each

---

[1]We use indices $0, \ldots, n-1$ rather than $1, \ldots, n$ only in this section, to simplify the discussion.

node will send two messages and receive two messages at every iteration, so the communication load is again balanced across the network.

**Undirected exponential graph.**    For the D-PSGD experiments we use a time-varying undirected bipartite exponential graph to represent the inter-node connectivity. Odd-numbered nodes send messages to peers that are $2^1 - 1, 2^2 - 1, \ldots, 2^{\lfloor \log_2(n-1) \rfloor} - 1$ (even-numbered nodes), and wait to a receive a message back in return. Each odd-numbered node cycles through the peers in the list in a similar fashion to the one-peer-per-node SGP experiments. Even-numbered nodes wait to receive a message from some peer (unknown to the receiving node ahead of time), and send a message back in return.

We adopt these graphs to be consistent with the experimental setup used in Lian et al. (2017) and Lian et al. (2018).

Note also that these graphs are all regular, in that all nodes have the same number of in-coming and out-going connections.

**Decentralized averaging errors.**    To further motivate our choice of using the directed exponential graph with SGP, let us forget about optimization for a moment and focus on the problem of distributed averaging, described in Section 2, using the PUSHSUM algorithm. Recall that each node $i$ starts with a vector $\boldsymbol{y}_i^{(0)}$, and the goal of the agents is to compute the average $\overline{\boldsymbol{y}} = \frac{1}{n} \sum_i \boldsymbol{y}_i^{(0)}$. Then, since $\boldsymbol{y}_i^{(k+1)} = \sum_{j=1}^n p_{i,j}^{(k)} \boldsymbol{y}_j^{(k)}$, after $k$ steps we have

$$\boldsymbol{Y}^{(k)} = \boldsymbol{P}^{(k-1)} \boldsymbol{P}^{(k-2)} \cdots \boldsymbol{P}^{(1)} \boldsymbol{P}^{(0)} \boldsymbol{Y}^{(0)},$$

where $\boldsymbol{Y}^{(k)}$ is a $n \times d$ matrix with $\boldsymbol{y}_i^{(k)}$ as its $i$th row.

Let $\boldsymbol{P}^{(k-1:0)} = \boldsymbol{P}^{(k-1)} \boldsymbol{P}^{(k-2)} \cdots \boldsymbol{P}^{(1)} \boldsymbol{P}^{(0)}$. The worst-case rate of convergence can be related to the second-largest singular value of $\boldsymbol{P}^{(k-1:0)}$ Nedić et al. (2018). In particular, after $k$ iterations we have

$$\sum_i \|\boldsymbol{y}_i^{(k)} - \overline{\boldsymbol{y}}\|_2^2 \leq \lambda_2(\boldsymbol{P}^{(k-1:0)}) \sum_i \|\boldsymbol{y}_i^{(0)} - \overline{\boldsymbol{y}}\|_2^2,$$

where $\lambda_2(\boldsymbol{P}^{(k-1:0)})$ denotes the second largest singular value of $\boldsymbol{P}^{(k-1:0)}$.

For the scheme proposed above, cycling deterministically through neighbors in the directed exponential graph, one can verify that after $k = \lfloor \log_2(n-1) \rfloor$ iterations, we have $\lambda_2(\boldsymbol{P}^{(k-1:0)}) = 0$, so all nodes exactly have the average. Intuitively, this happens because the directed exponential graph has excellent mixing properties: from any starting node in the network, one can get to any other node in at most $\log_2(n)$ hops. For $n = 32$ nodes, after 5 iterations averaging has converged using this strategy. In comparison, if one were to cycle through edges of the complete graph (where every node is connected to every other node), then for $n = 32$, after 5 consecutive iterations one would have still have $\lambda_2(\boldsymbol{P}^{(k-1:0)}) \approx 0.6$; i.e., nodes could be much further from the average (and hence, much less well-synchronized).

Similarly, one could consider designing the matrices $\boldsymbol{P}^{(k)}$ in a stochastic manner, where each node randomly samples one neighbor to send to at every iteration. If each node samples a destination uniformly from its set of neighbors in the directed exponential graph, then $\mathbb{E}\lambda_2(\boldsymbol{P}^{(k-1:0)}) \approx 0.4$, and if each node randomly selected a destination uniformly among all other nodes in the network (i.e., randomly from neighbors in the complete graph), then $\mathbb{E}\lambda_2(\boldsymbol{P}^{(k-1:0)}) \approx 0.2$. Thus, random schemes are still not as effective at quickly averaging as deterministically cycling through neighbors in the directed exponential graph. Moreover, with randomized schemes, we are no longer guaranteed that each node receives the same number of messages at every iteration, so the communication load will not be balanced as in the deterministic scheme.

The above discussion focused only on approximate distributed averaging, which is a key step within decentralized optimization. When averaging occurs less quickly, this also impacts optimization. Specifically, since nodes are less well-synchronized (i.e., further from a consensus), each node will be evaluating its local mini-batch gradient at a different point in parameter space. Averaging these points (rather than updates based on mini-batch gradients evaluated at the same point) can be seen as injecting additional noise into the optimization process, and in our experience this can lead to worse performance in terms of train and generalization errors.

## A.2 STOCHASTIC GRADIENT PUSH

In all of our experiments, we minimize the number of floating-point operations performed in each iteration, $k$, by using the mixing weights

$$p_{j,i}^{(k)} = 1/\left|\mathcal{N}_i^{\text{out}(k)}\right|$$

for all $i, j = 1, 2, \ldots, n$. In words, each node assigns mixing weights uniformly to all of its out-neighbors in each iteration. Recalling our convention that each node is an in- and out-neighbor of itself, it is easy to see that this choice of mixing-weight satisfies the column-stochasticity property. It may very well be that there is a different choice of mixing-weights that lead to better spectral properties of the gossip algorithm; however we leave this exploration for future work. We denote node $i$'s uniform mixing weights at time $t$ by $p_i^{(k)}$ — dropping the other subscript, which identifies the receiving node.

To maximize the utility of the resources available on each server, each node (occupying a single server exclusively) runs two threads, a gossip thread and a computation thread. The computation thread executes the main logic used to train the local model on the GPUs available to the node, while the communication thread is used for inter-node network I/O. In particular, the communication thread is used to gossip messages between nodes. When using Ethernet-based communication, the nodes communicate their parameter tensors over CPUs. When using InifiniBand-based communication, the nodes communicate their parameter tensors using GPUDirect RDMA, thereby avoiding superfluous device to pinned-memory transfers of the model parameters.

Each node initializes its model on one of its GPUs, and initializes its scalar push-sum weight to 1. At the start of training, each node also allocates a *send-* and a *receive-* communication-buffer in pinned memory on the CPU (or equivalently on a GPU in the case of GPUDirect RDMA communication).

In each iteration, the communication thread waits for the send-buffer to be filled by the computation thread; transmits the message in the send-buffer to its out-neighbours; and then aggregates any newly-received messages into the receive-buffer.

In each iteration, the computation thread blocks to retrieve the aggregated messages in the receive-buffer; directly adds the received parameters to its own model parameters; and directly adds the received push-sum weights to its own push-sum weight. The computation thread then converts the model parameters to the *de-biased* estimate by dividing by the push-sum weight; executes a forward-backward pass of the *de-biased model* in order to compute a stochastic mini-batch gradient; converts the model parameters back to the *biased estimate* by multiplying by the push-sum weight; and applies the newly-computed stochastic gradients to the biased model. The updated model parameters are then multiplied by the mixing weight, $p_i^{(k)}$, and asynchronously copied back into the send-buffer for use by the communication thread. The push-sum weight is also multiplied by the same mixing weight and concatenated into the send-buffer.

In short, gossip is performed on the biased model parameters (push-sum numerators); stochastic gradients are computed using the de-biased model parameters; stochastic gradients are applied back to the biased model parameters; and then the biased-model and the push-sum weight are multiplied by the same uniform mixing-weight and copied back into the send-buffer.

## A.3 HYPERPARAMETERS

When we "apply the stochastic gradients" to the biased model parameters, we actually carry out an SGD step with nesterov momentum. For the $32, 64$, and $128$ GPU experiments we use the same exact learning-rate, schedule, momentum, and weight decay as those suggested in (Goyal et al., 2017) for SGD. In particular, we use a reference learning-rate of $0.1$ with respect to a $256$ sample batch, and scale this linearly with the batch-size; we decay the learning-rate by a factor of $10$ at epochs $30, 60, 80$; we use a nesterov momentum parameter of $0.9$, and we use weight decay $0.0001$. For the $256$ GPU experiments, we decay the learning-rate by a factor of $10$ at epochs $40, 70, 85$, and we use a reference learning-rate of $0.0375$. In the $256$ GPU experiment with two peers-per-node, we revert to the original learning-rate and schedule.

---

**Algorithm 2** Stochastic Gradient Push with Momentum

---

**Require:** Initialize $\gamma > 0$, $m \in (0,1)$, $\boldsymbol{x}_i^{(0)} = \boldsymbol{z}_i^{(0)} \in \mathbb{R}^d$ and $w_i^{(0)} = 1$ for all nodes $i \in [n]$

1: **for** $k = 0, 1, 2, \cdots, K$ **do** at node $i$

2:      Sample new mini-batch $\xi_i^{(k)} \sim \mathcal{D}_i$ from local distribution

3:      Compute a local stochastic mini-batch gradient at $\boldsymbol{z}_i^{(k)}$: $\nabla \boldsymbol{F}_i(\boldsymbol{z}_i^{(k)}; \xi_i^{(k)})$

4:      $\boldsymbol{u}_i^{(k+1)} = m\boldsymbol{u}_i^{(k)} + \nabla \boldsymbol{F}_i(\boldsymbol{z}_i^{(k)}; \xi_i^{(k)})$

5:      $\boldsymbol{x}_i^{(k+\frac{1}{2})} = \boldsymbol{x}_i^{(k)} - \gamma(m\boldsymbol{u}_i^{(k+1)} + \nabla \boldsymbol{F}_i(\boldsymbol{z}_i^{(k)}; \xi_i^{(k)}))$

6:      Send $\left(p_{j,i}^{(k)}\boldsymbol{x}_i^{(k+\frac{1}{2})}, p_{j,i}^{(k)}w_i^{(k)}\right)$ to out-neighbors $j \in \mathcal{N}_i^{\text{out}(k)}$;

       receive $\left(p_{i,j}^{(k)}\boldsymbol{x}_j^{(k+\frac{1}{2})}, p_{i,j}^{(k)}w_j^{(k)}\right)$ from in-neighbors $j \in \mathcal{N}_i^{\text{in}(k)}$

7:      $\boldsymbol{x}_i^{(k+1)} = \sum_{j \in \mathcal{N}_i^{\text{in}(k)}} p_{i,j}^{(k)}\boldsymbol{x}_j^{(k+\frac{1}{2})}$

8:      $w_i^{(k+1)} = \sum_{j \in \mathcal{N}_i^{\text{in}(k)}} p_{i,j}^{(k)}w_j^{(k)}$

9:      $\boldsymbol{z}_i^{(k+1)} = \boldsymbol{x}_i^{(k+1)}/w_i^{(k+1)}$

10: **end for**

---

## B    Extra Experiments

### B.1    Additional Training Curves

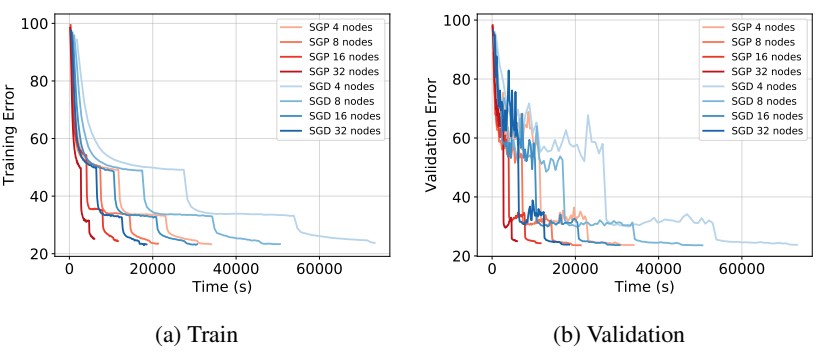

(a) Train                   (b) Validation

Figure B.1: Training on Ethernet 10Gbit/s

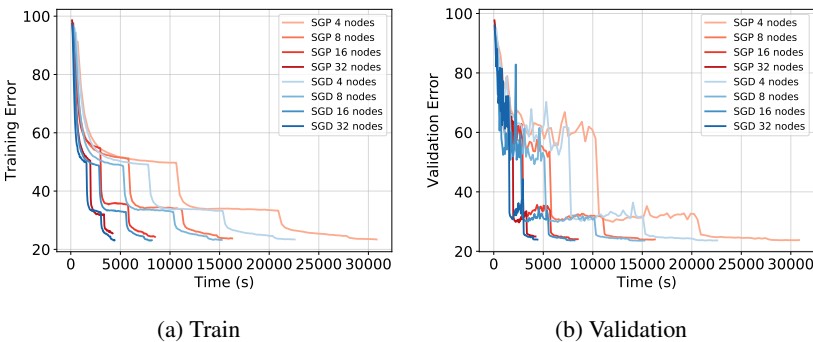

(a) Train                   (b) Validation

Figure B.2: Training on InfiniBand 100Gbit/s

FigureB.1 show the train and validation curve for the different runs performed on Ethernet 10Gbit/s. Figure B.2 show the train and validation curve for the different runs performed on InfiniBand 100Gbit/s.

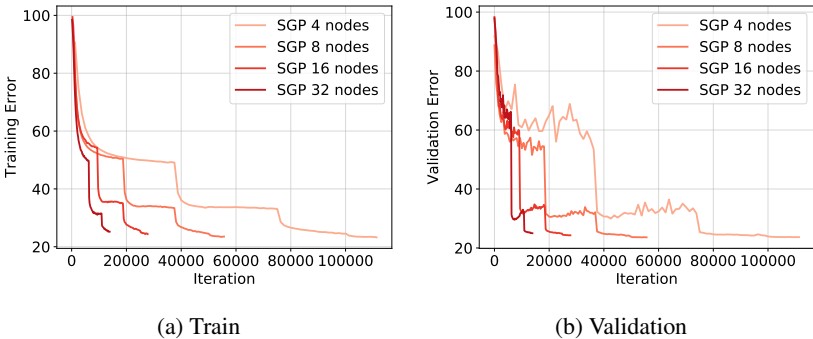

(a) Train

(b) Validation

Figure B.3: Training/Validation accuracy per iteration for SGP (Ethernet 10Gbit/s). Each time we double the number of node in the network, we half the total number of iterations.

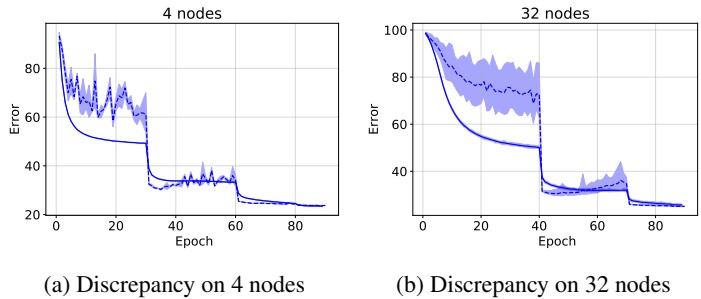

(a) Discrepancy on 4 nodes

(b) Discrepancy on 32 nodes

Figure B.4: Resnet50, trained with SGP, training and validation errors for 4 and 32 nodes experiments. The solid and dashed lines in each figure show the mean training and validation error, respectively, over all nodes. The shaded region shows the maximum and minimum error attained at different nodes in the same experiment. Although there is non-trivial variability across nodes early in training, all nodes eventually converge to similar validation errors, achieving consensus in the sense that they represent the same function.

Figure B.3 reports the training and validation accuracy of SGP when using a high-latency interconnect. As we scale up the number of nodes $n$, we scale down the total number of iterations $K$ to $K/n$ following Theorem 1. In particular, 32-node runs involves 8 times fewer global iterations than 4-node runs. We additionally report the total number of iterations and the final performances in Table 3. While we reduce the total number iterations by a factor of 8 when going from 4 to 32 nodes, the validation accuracy and training accuracy of the 32 node runs remain within 1.7% and 2.6%, respeively, of the validation and training accuracy achieved by the 4-node runs (and remains within the 1.2% of ALLREDUCE SGD accuracies).

| Nodes | 4 | 8 | 16 | 32 |
|---|---|---|---|---|
| Iterations | 112590 | 56250 | 28080 | 14040 |
| Training (%) | 76.75 | 76.59 | 75.64 | 74.79 |
| Validation (%) | 76.3 | 76.40 | 75.73 | 75.00 |

Table 3: Total number of iterations and final training and validation performances when training a Resnet50 on ImageNet using SGP over Ethernet 10Gbit/s.

## B.2 Discrepancy across different nodes

Here, we investigate the performance variability across nodes during training for SGP. In figure B.4, we report the minimum, maximum and mean error across the different nodes for training and vali-

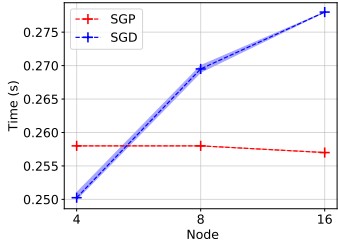

Figure B.5: Average time per training iteration for model trained on 4, 8 and 16 nodes using data copied on the local nodes and InfiniBand interconnect. SGP time per training iteration remains approximatelly constant as we increase the number of node, while SGD shows a slight increase.

dation. In an initial training phase, we observe that nodes have different validation errors; their local copies of the Resnet-50 model diverge. As we decrease the learning, the variability between the different nodes diminish and the nodes eventually converging to similar errors. This suggests that all models ultimately represent the same function, achieving consensus.

### B.3 TIMING ON INFINIBAND WITH LOCAL DATA COPY

To better isolate the effects of data-loading, we ran experiments on 32, 64, and 128 GPUs, where we first copied the data locally on every node. In that setting, we observe in Figure B.5 that the time-per-iteration of SGP remains approximately constant as we increase the number of nodes in the network, while the time for ALLREDUCE SGD increases.

### B.4 SGP SCALING ANALYSIS

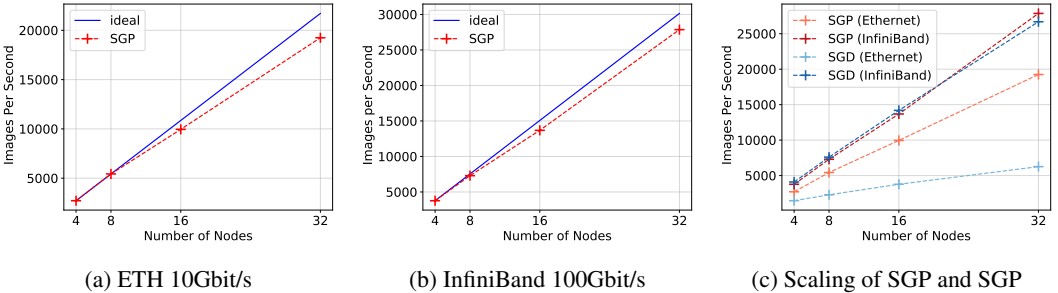

(a) ETH 10Gbit/s        (b) InfiniBand 100Gbit/s        (c) Scaling of SGP and SGP

Figure B.6: SGP throughput on Ethernet (a) and InfiniBand (b). SGP exhibits 88.6% scaling efficiency on Ethernet 10Gbit/s and 92.4% on InfiniBand. Comparison of SGD vs SGP throughput in Figure (c) shows that SGP exhibit better scaling and is more robust to high-latency interconnect.

Figure B.6 highlights SGP input images throughput as we scale up the number of cluster node on both Ethernet 10Gbit/s and Infiniband 100Gbit/s. SGP exhibits 88.6% scaling efficiency on Ethernet 10Gbit/s and 92.4% on InfiniBand and stay close to the ideal scaling in both cases. In addition Figure (c) shows that SGP exhibit better scaling as we increase the network size and is more robust to high-latency interconnect.

## C PROOFS OF THEORETICAL GUARANTEES

Our convergence rate analysis is divided into three main parts. In the first one (subsection C.1) we present upper bounds for three important expressions that appear in our computations. In subsection C.2 we focus on proving the important for our analysis Lemma 8 based on which we later build the

proofs of our main Theorems. Finally in the third part (subsection C.3) we provide the proofs for Theorems 1 and 2.

**Preliminary results.**    In our analysis two preliminary results are extensively used. We state them here for future reference.

- Let $a, b \in \mathbb{R}$. Since $(a - b)^2 \geq 0$, it holds that

$$2ab \leq a^2 + b^2. \tag{7}$$

 Thus, $\|\boldsymbol{x}\| \|\boldsymbol{y}\| \leq (\|\boldsymbol{x}\|^2 + \|\boldsymbol{y}\|^2)/2$.
- Let $r \in (0, 1)$ then from the summation of geometric sequence and for any $K \leq \infty$ it holds that

$$\sum_{k=0}^{K} r^k \leq \sum_{k=0}^{\infty} r^k = \frac{1}{1 - r} \tag{8}$$

**Matrix Representation.**    The presentation of stochastic gradient push (Algorithm 1) was done from node i's perspective for all $i \in [n]$. Note however, that the update rule of SGP at the $k^{th}$ iteration can be viewed from a global viewpoint. To see this let us define the following matrices (concatenation of the values of all nodes at the $k^{th}$ iteration):

$$\mathbf{X}^{(k)} = \left[ \boldsymbol{x}_1^{(k)}, \boldsymbol{x}_2^{(k)}, \ldots, \boldsymbol{x}_n^{(k)} \right] \in \mathbb{R}^{d \times n} \quad \xi^{(k)} = \left[ \xi_1^{(k)}, \xi_2^{(k)}, \ldots, \xi_n^{(k)} \right] \in \mathbb{R}^n$$

$$\mathbf{Z}^{(k)} = \left[ \boldsymbol{z}_1^{(k)}, \boldsymbol{z}_2^{(k)}, \ldots, \boldsymbol{z}_n^{(k)} \right] \in \mathbb{R}^{d \times n}$$

$$\nabla F(\mathbf{Z}^{(k)}, \xi^{(k)}) = \left[ \nabla F_1(\boldsymbol{z}_1^{(k)}; \xi_1^{(k)}), \nabla F_2(\boldsymbol{z}_2^{(k)}; \xi_2^{(k)}), \ldots, \nabla F_n(\boldsymbol{z}_n^{(k)}; \xi_n^{(k)}) \right] \in \mathbb{R}^{d \times n}$$

$$\nabla f(\mathbf{Z}^{(k)}) = \left[ \nabla f_1(\boldsymbol{z}_1^{(k)}), \nabla f_2(\boldsymbol{z}_2^{(k)}), \ldots, \nabla f_n(\boldsymbol{z}_n^{(k)}) \right] \in \mathbb{R}^{d \times n}$$

Using the above matrices, the $6^{th}$ step of SGP (Algorithm 1) can be expressed as follows [2]:

$$\mathbf{X}^{(k+1)} = \left( \mathbf{X}^{(k)} - \gamma \nabla F(\mathbf{Z}^{(k)}, \xi^{(k)}) \right) [\boldsymbol{P}^{(k)}]^T \tag{9}$$

where $[\boldsymbol{P}^{(k)}]^T$ is the transpose of matrix $\boldsymbol{P}^k$ with entries:

$$p_{i,j}^{(k)} = \begin{cases} 1/d_j(k), & \text{if } j \in \mathcal{N}_i^{\text{in}(k)}. \\ 0, & \text{otherwise.} \end{cases} \tag{10}$$

Recall that we also have $\overline{\boldsymbol{x}}^{(k)} = \frac{\mathbf{X}^{(k)} \mathbf{1}_n}{n} = \frac{1}{n} \sum_{i=1}^{n} x_i^{(k)}$.

**Bound for the mixing matrices.**    Next we state a known result from the control literature studying consensus-based optimization which allows us to bound the distance between the de-biased parameters at each node and the node-wise average.

Recall that we have assumed that the sequence of mixing matrices $\boldsymbol{P}^{(k)}$ are $B$-strongly connected. A directed graph is called *strongly connected* if every pair of vertices is connected with a directed path (*i.e.*, following the direction of edges), and the $B$-strongly connected assumption is that the graph with edge set $\bigcup_{k=lB}^{(l+1)B-1} E^{(k)}$ is strongly connected, for every $l \geq 0$.

We have also assumed that for all $k \geq 0$, each column of $\boldsymbol{P}^{(k)}$ has $D$ non-zero entries, and the diameter of the graph with edge set $\bigcup_{k=lB}^{(l+1)B-1} E^{(k)}$ has diameter at most $\Delta$. Based on these assumptions, after $\Delta B$ consecutive iterations, the product

$$\boldsymbol{A}^{(k)} := \boldsymbol{P}^{(k+\Delta B-1)} \ldots \boldsymbol{P}^{(k+1)} \boldsymbol{P}^{(k)}$$

has no non-zero entries. Moreover, every entry of $\boldsymbol{A}^{(k)}$ is at least $D^{-\Delta B}$.

---

[2] Note that in a similar way we can obtain matrix expressions for steps 7 and 8 of Algorithm 1.

**Lemma 3.** *Suppose that Assumption 3 (mixing connectivity) holds. Let $\lambda = 1 - nD^{-\Delta B}$ and let $q = \lambda^{1/(\Delta B + 1)}$. Then there exists a constant*

$$C < \frac{2\sqrt{d}D^{\Delta B}}{\lambda^{\frac{\Delta B + 2}{\Delta B + 1}}},$$

*where $d$ is the dimension of $\overline{\boldsymbol{x}}^{(k)}$, $z_i^{(k)}$, and $x_i^{(0)}$, such that, for all $i = 1, 2, \ldots, n$ and $k \geq 0$,*

$$\left\| \overline{\boldsymbol{x}}^{(k)} - z_i^{(k)} \right\|_2^2 \leq \left( Cq^k \left\| x_i^{(0)} \right\|_2 + \gamma C \sum_{s=0}^{k} q^{k-s} \left\| \nabla F_i(z_i^{(s)}; \xi_i^{(s)}) \right\|_2 \right)^2.$$

This particular lemma follows after a small adaptation to Theorem 1 in Assran & Rabbat (2018) and its proof is based on Wolfowitz (1963). Similar bounds appear in a variety of other papers, including Nedić & Olshevsky (2016).

## C.1 IMPORTANT UPPER BOUNDS

**Lemma 4** (Bound of stochastic gradient). *We have the following inequality under Assumptions 1 and 2:*

$$\mathbb{E} \left\| \nabla f_i(\boldsymbol{z}_i^{(k)}) \right\|^2 \leq 3L^2 \mathbb{E} \left\| \boldsymbol{z}_i^{(k)} - \overline{\boldsymbol{x}}^{(k)} \right\|^2 + 3\zeta^2 + 3\mathbb{E} \left\| \nabla f(\overline{\boldsymbol{x}}^{(k)}) \right\|^2$$

*Proof.*

$$
\begin{aligned}
\mathbb{E} \left\| \nabla f_i(\boldsymbol{z}_i^{(k)}) \right\|^2 \quad &\leq \quad 3\mathbb{E} \left\| \nabla f_i(\boldsymbol{z}_i^{(k)}) - \nabla f_i(\overline{\boldsymbol{x}}^{(k)}) \right\|^2 + 3\mathbb{E} \left\| \nabla f_i(\overline{\boldsymbol{x}}^{(k)}) - \nabla f(\overline{\boldsymbol{x}}^{(k)}) \right\|^2 + 3\mathbb{E} \left\| \nabla f(\overline{\boldsymbol{x}}^{(k)}) \right\|^2 \\
&\overset{\text{L-smooth}}{\leq} \quad 3L^2 \mathbb{E} \left\| \boldsymbol{z}_i^{(k)} - \overline{\boldsymbol{x}}^{(k)} \right\|^2 + 3\mathbb{E} \left\| \nabla f_i(\overline{\boldsymbol{x}}^{(k)}) - \nabla f(\overline{\boldsymbol{x}}^{(k)}) \right\|^2 + 3\mathbb{E} \left\| \nabla f(\overline{\boldsymbol{x}}^{(k)}) \right\|^2 \\
&\overset{\text{Bounded Variance}}{\leq} \quad 3L^2 \mathbb{E} \left\| \boldsymbol{z}_i^{(k)} - \overline{\boldsymbol{x}}^{(k)} \right\|^2 + 3\zeta^2 + 3\mathbb{E} \left\| \nabla f(\overline{\boldsymbol{x}}^{(k)}) \right\|^2
\end{aligned}
\tag{11}
$$

$\square$

**Lemma 5.** *Let Assumptions 1-3 hold. Then,*

$$
\begin{aligned}
Q_i^{(k)} = \mathbb{E} \left\| \overline{\boldsymbol{x}}^{(k)} - z_i^{(k)} \right\|^2 \quad \leq \quad & \left( \gamma^2 \frac{4C^2}{(1-q)^2} + \gamma \frac{q^k C^2}{1-q} \right) \sigma^2 + \left( \gamma^2 \frac{12C^2}{(1-q)^2} + \gamma \frac{q^k 3C^2}{1-q} \right) \zeta^2 \\
+ \quad & \left( \gamma^2 \frac{12L^2 C^2}{1-q} + \gamma q^k 3L^2 C^2 \right) \sum_{j=0}^{k} q^{k-j} Q_i^{(j)} \\
+ \quad & \left( \gamma^2 \frac{12C^2}{1-q} + \gamma q^k 3C^2 \right) \sum_{j=0}^{k} q^{k-j} \mathbb{E} \left\| \nabla f(\overline{\boldsymbol{x}}^{(j)}) \right\|^2 \\
+ \quad & \left( q^{2k} C^2 + \gamma q^k \frac{2C^2}{1-q} \right) \left\| \boldsymbol{x}_i^{(0)} \right\|^2.
\end{aligned}
\tag{12}
$$

*Proof.*

$$
\begin{aligned}
Q_i^{(k)} \quad &= \quad \mathbb{E} \left\| \overline{\boldsymbol{x}}^{(k)} - \boldsymbol{z}_i{}^{(k)} \right\|^2 \\
&\overset{Lemma\ 3}{\leq} \quad \mathbb{E} \left( Cq^k \left\| \boldsymbol{x}_i{}^{(0)} \right\| + \gamma C \sum_{s=0}^{k} q^{k-s} \left\| \nabla F_i(\boldsymbol{z}_i^{(s)}; \xi_i^{(s)}) \right\| \right)^2 \\
&= \quad \mathbb{E} \left( Cq^k \left\| \boldsymbol{x}_i{}^{(0)} \right\| + \gamma C \sum_{s=0}^{k} q^{k-s} \left\| \nabla F_i(\boldsymbol{z}_i^{(s)}; \xi_i^{(s)}) - \nabla f_i(\boldsymbol{z}_i^{(s)}) + \nabla f_i(\boldsymbol{z}_i^{(s)}) \right\| \right)^2 \\
&\leq \quad \mathbb{E} \left( \underbrace{Cq^k \left\| \boldsymbol{x}_i{}^{(0)} \right\|}_{a} + \underbrace{\gamma C \sum_{s=0}^{k} q^{k-s} \left\| \nabla F_i(\boldsymbol{z}_i^{(s)}; \xi_i^{(s)}) - \nabla f_i(\boldsymbol{z}^{(s)}) \right\|}_{b} + \underbrace{\gamma C \sum_{s=0}^{k} q^{k-s} \left\| \nabla f_i(\boldsymbol{z}_i^{(s)}) \right\|}_{c} \right)^2
\end{aligned}
$$
(13)

Thus, using the above expressions of $a$, $b$ and $c$ we have that $Q_i^{(k)} \leq \mathbb{E}(a^2 + b^2 + c^2 + 2ab + 2bc + 2ac)$. Let us now obtain bounds for all of these quantities:

$$
a^2 \quad = \quad C^2 \left\| \boldsymbol{x}_i{}^{(0)} \right\|^2 q^{2k}
$$

$$
\begin{aligned}
b^2 \quad = \quad &\gamma^2 C^2 \sum_{j=0}^{k} q^{2(k-j)} \left\| \nabla F_i(\boldsymbol{z}_i^{(j)}; \xi_i^{(j)}) - \nabla f_i(\boldsymbol{z}_i^{(j)}) \right\|^2 \\
&+ \underbrace{2\gamma^2 C^2 \sum_{j=0}^{k} \sum_{s=j+1}^{k} q^{2k-j-s} \left\| \nabla F_i(\boldsymbol{z}_i^{(j)}; \xi_i^{(j)}) - \nabla f_i(\boldsymbol{z}_i^{(j)}) \right\| \left\| \nabla F_i(\boldsymbol{z}_i^{(s)}; \xi_i^{(s)}) - \nabla f_i(\boldsymbol{z}_i^{(s)}) \right\|}_{b_1}
\end{aligned}
$$

$$
c^2 \quad = \quad \gamma^2 C^2 \sum_{j=0}^{k} q^{2(k-j)} \left\| \nabla f_i(\boldsymbol{z}_i^{(j)}) \right\|^2 + \underbrace{2\gamma^2 C^2 \sum_{j=0}^{k} \sum_{s=j+1}^{k} q^{2k-j-s} \left\| \nabla f_i(\boldsymbol{z}_i^{(j)}) \right\| \left\| \nabla f_i(\boldsymbol{z}_i^{(s)}) \right\|}_{c_1}
$$

$$
2ab \quad = \quad 2\gamma C^2 q^k \left\| \boldsymbol{x}_i{}^{(0)} \right\| \sum_{s=0}^{k} q^{k-s} \left\| \nabla F_i(\boldsymbol{z}_i^{(s)}; \xi_i^{(s)}) - \nabla f_i(\boldsymbol{z}_i^{(s)}) \right\|
$$

$$
2ac \quad = \quad 2\gamma C^2 q^k \left\| \boldsymbol{x}_i{}^{(0)} \right\| \sum_{s=0}^{k} q^{k-s} \left\| \nabla f_i(\boldsymbol{z}_i^{(s)}) \right\|
$$

$$
2bc \quad = \quad 2\gamma^2 C^2 \sum_{j=0}^{k} \sum_{s=0}^{k} q^{2k-j-s} \left\| \nabla F_i(\boldsymbol{z}_i^{(j)}; \xi_i^{(j)}) - \nabla f_i(\boldsymbol{z}_i^{(j)}) \right\| \left\| \nabla f_i(\boldsymbol{z}_i^{(s)}) \right\|.
$$

The expression $b_1$ is bounded as follows:

$$
\begin{aligned}
b_1 \;=\;& \gamma^2 C^2 \sum_{j=0}^{k}\sum_{s=j+1}^{k} q^{2k-j-s} 2 \left\| \nabla F_i(z_i^{(j)};\xi_i^{(j)}) - \nabla f_i(z_i^{(j)}) \right\| \left\| \nabla F_i(z_i^{(s)};\xi_i^{(s)}) - \nabla f_i(z_i^{(s)}) \right\| \\
\stackrel{(7)}{\leq}\;& \gamma^2 C^2 \sum_{j=0}^{k}\sum_{s=j+1}^{k} q^{2k-s-j} \left\| \nabla F_i(z_i^{(j)};\xi_i^{(j)}) - \nabla f_i(z_i^{(j)}) \right\|^2 \\
+\;& \gamma^2 C^2 \sum_{j=0}^{k}\sum_{s=j+1}^{k} q^{2k-s-j} \left\| \nabla F_i(z_i^{(s)};\xi_i^{(s)}) - \nabla f_i(z_i^{(s)}) \right\|^2 \\
\leq\;& \gamma^2 C^2 \sum_{j=0}^{k}\sum_{s=0}^{k} q^{2k-s-j} \left\| \nabla F_i(z_i^{(j)};\xi_i^{(j)}) - \nabla f_i(z_i^{(j)}) \right\|^2 \\
+\;& \gamma^2 C^2 \sum_{j=0}^{k}\sum_{s=0}^{k} q^{2k-s-j} \left\| \nabla F_i(z_i^{(s)};\xi_i^{(s)}) - \nabla f_i(z_i^{(s)}) \right\|^2 \\
=\;& \gamma^2 C^2 \sum_{j=0}^{k} q^{k-j} \left\| \nabla F_i(z_i^{(j)};\xi_i^{(j)}) - \nabla f_i(z_i^{(j)}) \right\|^2 \sum_{s=0}^{k} q^{k-s} \\
+\;& \gamma^2 C^2 \sum_{s=0}^{k} q^{k-s} \left\| \nabla F_i(z_i^{(s)};\xi_i^{(s)}) - \nabla f_i(z_i^{(s)}) \right\|^2 \sum_{j=0}^{k} q^{k-j} \\
\stackrel{(8)}{\leq}\;& \frac{1}{1-q}\gamma^2 C^2 \sum_{j=0}^{k} q^{k-j} \left\| \nabla F_i(z_i^{(j)};\xi_i^{(j)}) - \nabla f_i(z_i^{(j)}) \right\|^2 \\
+\;& \frac{1}{1-q}\gamma^2 C^2 \sum_{s=0}^{k} q^{k-s} \left\| \nabla F_i(z_i^{(s)};\xi_i^{(s)}) - \nabla f_i(z_i^{(s)}) \right\|^2 \\
=\;& \frac{2}{1-q}\gamma^2 C^2 \sum_{j=0}^{k} q^{k-j} \left\| \nabla F_i(z_i^{(j)};\xi_i^{(j)}) - \nabla f_i(z_i^{(j)}) \right\|^2 .
\end{aligned}
\tag{14}
$$

Thus,

$$
\begin{aligned}
b^2 \;=\;& \gamma^2 C^2 \sum_{j=0}^{k} q^{2(k-j)} \left\| \nabla F_i(z_i^{(j)};\xi_i^{(j)}) - \nabla f_i(z_i^{(j)}) \right\|^2 + b_1 \\
\leq\;& \frac{\gamma^2 C^2}{1-q} \sum_{j=0}^{k} q^{k-j} \left\| \nabla F_i(z_i^{(j)};\xi_i^{(j)}) - \nabla f_i(z_i^{(j)}) \right\|^2 + b_1 \\
\stackrel{(14)}{\leq}\;& \frac{3\gamma^2 C^2}{1-q} \sum_{j=0}^{k} q^{k-j} \left\| \nabla F_i(z_i^{(j)};\xi_i^{(j)}) - \nabla f_i(z_i^{(j)}) \right\|^2
\end{aligned}
\tag{15}
$$

where in the first inequality above we use the fact that for $q \in (0,1)$, we have $q^k < \frac{1}{1-q}, \forall k > 0$.
By identical construction we have

$$
c^2 \leq \frac{3\gamma^2 C^2}{1-q} \sum_{j=0}^{k} q^{k-j} \left\| \nabla f_i(z_i^{(j)}) \right\|^2 .
$$

Now let us bound the products $2ab$, $2ac$ and $2bc$.

$$
\begin{aligned}
2ab &= \gamma C^2 q^k \sum_{s=0}^{k} q^{k-s} 2 \left\| \boldsymbol{x_i}^{(0)} \right\| \left\| \nabla F_i(\boldsymbol{z}_i^{(s)}; \xi_i^{(s)}) - \nabla f_i(\boldsymbol{z}_i^{(s)}) \right\| \\
&\overset{(7)}{\leq} \gamma C^2 q^k \sum_{j=0}^{k} q^{k-j} \left\| \nabla F_i(\boldsymbol{z}_i^{(j)}; \xi_i^{(j)}) - \nabla f_i(\boldsymbol{z}_i^{(j)}) \right\|^2 + \gamma C^2 q^k \sum_{j=0}^{k} q^{k-j} \left\| \boldsymbol{x_i}^{(0)} \right\|^2 \\
&\overset{(8)}{\leq} \gamma C^2 q^k \sum_{j=0}^{k} q^{k-j} \left\| \nabla F_i(\boldsymbol{z}_i^{(j)}; \xi_i^{(j)}) - \nabla f_i(\boldsymbol{z}_i^{(j)}) \right\|^2 + \frac{\gamma C^2 \left\| x_i^{(0)} \right\|^2}{1-q} q^k
\end{aligned}
\tag{16}
$$

By similar procedure,

$$
2ac \leq \gamma C^2 q^k \sum_{s=0}^{k} q^{k-s} \left\| \nabla f_i(z_i^{(s)}) \right\|^2 + \frac{\gamma C^2 \left\| x_i^{(0)} \right\|^2}{1-q} q^k
\tag{17}
$$

Finally,

$$
\begin{aligned}
2bc &= \gamma^2 C^2 \sum_{j=0}^{k} \sum_{s=0}^{k} q^{2k-j-s} 2 \left\| \nabla F_i(z_i^{(j)}; \xi_i^{(j)}) - \nabla f_i(z_i^{(j)}) \right\| \left\| \nabla f_i(z_i^{(s)}) \right\| \\
&\overset{(7)}{\leq} \gamma^2 C^2 \sum_{j=0}^{k} \sum_{s=0}^{k} q^{2k-j-s} \left\| \nabla F_i(z_i^{(j)}; \xi_i^{(j)}) - \nabla f_i(z_i^{(j)}) \right\|^2 + \gamma^2 C^2 \sum_{j=0}^{k} \sum_{s=0}^{k} q^{2k-j-s} \left\| \nabla f_i(z_i^{(s)}) \right\|^2, \\
&= \gamma^2 C^2 \sum_{j=0}^{k} q^{k-j} \left\| \nabla F_i(z_i^{(j)}; \xi_i^{(j)}) - \nabla f_i(z_i^{(j)}) \right\|^2 \sum_{s=0}^{k} q^{k-s} + \gamma^2 C^2 \sum_{s=0}^{k} q^{k-s} \left\| \nabla f_i(z_i^{(s)}) \right\|^2 \sum_{j=0}^{k} q^{k-j}, \\
&\overset{(8)}{\leq} \frac{\gamma^2 C^2}{1-q} \sum_{j=0}^{k} q^{k-j} \left\| \nabla F_i(z_i^{(j)}; \xi_i^{(j)}) - \nabla f_i(z_i^{(j)}) \right\|^2 + \frac{\gamma^2 C^2}{1-q} \sum_{s=0}^{k} q^{k-s} \left\| \nabla f_i(z_i^{(s)}) \right\|^2
\end{aligned}
\tag{18}
$$

By combining all of the above bounds together we obtain:

$$
\begin{aligned}
Q_i^{(k)} &\leq \mathbb{E}(a^2 + b^2 + c^2 + 2ab + 2bc + 2ac) \\
&\leq \mathbb{E} \frac{4\gamma^2 C^2}{1-q} \sum_{j=0}^{k} q^{k-j} \left\| \nabla F_i(z_i^{(j)}; \xi_i^{(j)}) - \nabla f_i(z_i^{(j)}) \right\|^2 \\
&+ \mathbb{E} \frac{4\gamma^2 C^2}{1-q} \sum_{j=0}^{k} q^{k-j} \left\| \nabla f_i(z_i^{(j)}) \right\|^2 \\
&+ C^2 \left\| x_i^{(0)} \right\|^2 q^{2k} \\
&+ \frac{2\gamma C^2 \left\| x_i^{(0)} \right\|^2}{1-q} q^k \\
&+ \mathbb{E} \gamma C^2 q^k \sum_{j=0}^{k} q^{k-j} \left\| \nabla f_i(z_i^{(j)}) \right\|^2 \\
&+ \mathbb{E} \gamma C^2 q^k \sum_{j=0}^{k} q^{k-j} \left\| \nabla F_i(z_i^{(j)}; \xi_i^{(j)}) - \nabla f_i(z_i^{(j)}) \right\|^2 .
\end{aligned}
\tag{19}
$$

After grouping terms together and using the upper bound of Lemma 4, we obtain

$$
\begin{aligned}
Q_i^{(k)} \quad &\leq \quad \left(\gamma^2 \frac{4C^2}{(1-q)^2} + \gamma \frac{q^k C^2}{1-q}\right)\sigma^2 + \left(q^{2k}C^2 + \gamma q^k \frac{2C^2}{1-q}\right)\left\|x_i^{(0)}\right\|^2 . \\
&+ \quad \left(\gamma^2 \frac{4C^2}{1-q} + \gamma q^k C^2\right)\sum_{j=0}^{k} q^{k-j}\mathbb{E}\left\|\nabla f_i(z_i^{(j)})\right\|^2 \\
&\overset{Lemma\ 4}{\leq} \quad \left(\gamma^2 \frac{4C^2}{(1-q)^2} + \gamma\frac{q^k C^2}{1-q}\right)\sigma^2 + \left(q^{2k}C^2 + \gamma q^k \frac{2C^2}{1-q}\right)\left\|x_i^{(0)}\right\|^2 \\
&+ \quad \left(\gamma^2 \frac{12C^2}{(1-q)^2} + \frac{\gamma q^k 3C^2}{1-q}\right)\zeta^2 \\
&+ \quad \left(\gamma^2 \frac{12L^2C^2}{1-q} + \gamma q^k 3L^2C^2\right)\sum_{j=0}^{k} q^{k-j}Q_i^{(j)} \\
&+ \quad \left(\gamma^2 \frac{12C^2}{1-q} + \gamma q^k 3C^2\right)\sum_{j=0}^{k} q^{k-j}\mathbb{E}\left\|\nabla f(\overline{x}^{(j)})\right\|^2 \quad (20)
\end{aligned}
$$

This completes the proof. □

Having found a bound for the quantity $Q_i^{(k)}$, let us know present a lemma for bounding the quantity $\sum_{k=0}^{K-1} M^{(k)}$ where $K > 1$ is a constant and $M^{(k)}$ is the average of $Q_i^{(k)}$ across all nodes $i \in [n]$. That is, $M^{(k)} = \frac{1}{n}\sum_{i=1}^{n} Q_i^{(k)}$.

**Lemma 6.** *Let Assumptions 1-3 hold and let us define $D_2 = 1 - \frac{\gamma^2 12L^2C^2}{(1-q)^2} - \frac{\gamma 3L^2C^2}{(1-q)^2}$ . Then,*

$$
\begin{aligned}
\sum_{k=0}^{K-1} M^{(k)} \quad &\leq \quad \left(\gamma^2 \frac{4C^2}{(1-q)^2 D_2}\right)\sigma^2 K + \left(\gamma\frac{C^2}{(1-q)^2 D_2}\right)\sigma^2 \\
&+ \left(\gamma^2 \frac{12C^2}{(1-q)^2 D_2}\right)\zeta^2 K + \left(\frac{\gamma 3C^2}{(1-q)^2 D_2}\right)\zeta^2 \\
&+ \left(\frac{C^2}{(1-q)^2 D_2} + \gamma\frac{2C^2}{(1-q)^2 D_2}\right)\frac{\sum_{i=1}^{n}\left\|x_i^{(0)}\right\|^2}{n} \\
&+ \left(\gamma^2 \frac{12C^2}{(1-q)^2 D_2} + \gamma\frac{3C^2}{(1-q)^2 D_2}\right)\sum_{k=0}^{K-1}\mathbb{E}\left\|\nabla f(\overline{x}^{(k)})\right\|^2 \quad (21)
\end{aligned}
$$

*Proof.* Using the bound for $Q_i^{(k)}$ let us first bound its average across all nodes $M^{(k)}$

$$
\begin{aligned}
M^{(k)} \quad &= \quad \frac{1}{n}\sum_{i=1}^{n} Q_i^{(k)} \\
&\overset{Lemma\ 5}{\leq} \quad \left(\gamma^2 \frac{4C^2}{(1-q)^2} + \gamma\frac{q^k C^2}{1-q}\right)\sigma^2 + \left(\gamma^2 \frac{12C^2}{(1-q)^2} + \frac{\gamma q^k 3C^2}{1-q}\right)\zeta^2 \\
&+ \quad \left(\gamma^2 \frac{12C^2}{1-q} + \gamma q^k 3C^2\right)\sum_{j=0}^{k} q^{k-j}\mathbb{E}\left\|\nabla f(\overline{x}^{(j)})\right\|^2 \\
&+ \quad \left(\gamma^2 \frac{12L^2C^2}{1-q} + \gamma q^k 3L^2C^2\right)\sum_{j=0}^{k} q^{k-j}M^{(j)} \\
&+ \quad \left(q^{2k}C^2 + \gamma q^k \frac{2C^2}{1-q}\right)\frac{\sum_{i=1}^{n}\left\|x_i^{(0)}\right\|^2}{n} . \quad (22)
\end{aligned}
$$

At this point note that for any $\lambda \in (0,1)$, non-negative integer $K \in \mathbb{N}$, and non-negative sequence $\{\beta^{(j)}\}_{j=0}^{k}$, it holds that

$$
\begin{aligned}
\sum_{k=0}^{K} \sum_{j=0}^{k} \lambda^{k-j} \beta^{(j)} &= \beta^{(0)} \left(\lambda^K + \lambda^{K-1} + \cdots + \lambda^0\right) + \beta^{(1)} \left(\lambda^{K-1} + \lambda^{K-2} + \cdots + \lambda^0\right) + \cdots + \beta^{(K)} \left(\lambda^0\right) \\
&\leq \frac{1}{1-\lambda} \sum_{j=0}^{K} \beta^{(j)}.
\end{aligned}
\tag{23}
$$

Similarly,

$$
\sum_{k=0}^{K} \lambda^k \sum_{j=0}^{k} \lambda^{k-j} \beta^{(j)} = \sum_{k=0}^{K} \sum_{j=0}^{k} \lambda^{2k-j} \beta^{(j)} \leq \sum_{k=0}^{K} \sum_{j=0}^{k} \lambda^{2(k-j)} \beta^{(j)} \overset{(23)}{\leq} \frac{1}{1-\lambda^2} \sum_{j=0}^{K} \beta^{(j)} \tag{24}
$$

Now by summing from $k = 0$ to $K - 1$ and using the bounds of (23) and (24) we obtain:

$$
\begin{aligned}
\sum_{k=0}^{K-1} M^{(k)} \leq & \left(\gamma^2 \frac{4C^2}{(1-q)^2}\right) \sigma^2 K + \left(\gamma \frac{C^2}{(1-q)^2}\right) \sigma^2 \\
& + \left(\gamma^2 \frac{12C^2}{(1-q)^2}\right) \zeta^2 K + \left(\frac{\gamma 3C^2}{1-q}\right) \zeta^2 \\
& + \left(\frac{C^2}{1-q^2} + \gamma \frac{2C^2}{(1-q)^2}\right) \frac{\sum_{i=1}^{n} \left\|x_i^{(0)}\right\|^2}{n} \\
& + \left(\gamma^2 \frac{12C^2}{(1-q)^2} + \gamma \frac{3C^2}{1-q^2}\right) \sum_{k=0}^{K-1} \mathbb{E}\left\|\nabla f(\overline{x}^{(k)})\right\|^2 \\
& + \left(\gamma^2 \frac{12L^2C^2}{(1-q)^2} + \gamma \frac{3L^2C^2}{1-q^2}\right) \sum_{k=0}^{K-1} M^{(k)}.
\end{aligned}
$$

By rearranging:

$$
\begin{aligned}
\left(1 - \gamma^2 \frac{12L^2C^2}{(1-q)^2} - \gamma \frac{3L^2C^2}{1-q^2}\right) \sum_{k=0}^{K-1} M^{(k)} \leq & \left(\gamma^2 \frac{4C^2}{(1-q)^2}\right) \sigma^2 K + \left(\gamma \frac{C^2}{(1-q)^2}\right) \sigma^2 \\
& + \left(\gamma^2 \frac{12C^2}{(1-q)^2}\right) \zeta^2 K + \left(\frac{\gamma 3C^2}{(1-q)^2}\right) \zeta^2 \\
& + \left(\frac{C^2}{1-q^2} + \gamma \frac{2C^2}{(1-q)^2}\right) \frac{\sum_{i=1}^{n} \left\|x_i^{(0)}\right\|^2}{n} \\
& + \left(\gamma^2 \frac{12C^2}{(1-q)^2} + \gamma \frac{3C^2}{1-q^2}\right) \sum_{k=0}^{K-1} \mathbb{E}\left\|\nabla f(\overline{x}^{(k)})\right\|^2
\end{aligned}
$$

Note that since $q \in (0,1)$ it holds that $\frac{1}{1-q^2} \leq \frac{1}{(1-q)^2}$.[3] Thus,

---

[3]This step is used to simplified the expressions involve the parameter $q$. One can still obtain similar results by keeping the expression $\frac{1}{1-q^2}$ in the definition of $D_2$

$$\left(1 - \gamma^2 \frac{12L^2C^2}{(1-q)^2} - \gamma \frac{3L^2C^2}{(1-q)^2}\right) \sum_{k=0}^{K-1} M^{(k)} \leq \left(\gamma^2 \frac{4C^2}{(1-q)^2}\right) \sigma^2 K + \left(\gamma \frac{C^2}{(1-q)^2}\right) \sigma^2$$

$$+ \left(\gamma^2 \frac{12C^2}{(1-q)^2}\right) \zeta^2 K + \left(\frac{\gamma 3C^2}{(1-q)^2}\right) \zeta^2$$

$$+ \left(\frac{C^2}{(1-q)^2} + \gamma \frac{2C^2}{(1-q)^2}\right) \frac{\sum_{i=1}^{n} \left\|x_i^{(0)}\right\|^2}{n}$$

$$+ \left(\gamma^2 \frac{12C^2}{(1-q)^2} + \gamma \frac{3C^2}{(1-q)^2}\right) \sum_{k=0}^{K-1} \mathbb{E}\left\|\nabla f(\overline{\boldsymbol{x}}^{(k)})\right\|^2$$

Dividing both sides with $D_2 = 1 - \frac{\gamma^2 12L^2C^2}{(1-q)^2} - \frac{\gamma 3L^2C^2}{(1-q)^2}$ completes the proof. $\qquad \square$

### C.2 Towards the proof of the main Theorems

The goal of this section is the presentation of Lemma 8. It is the main lemma of our convergence analysis and based on which we build the proofs of Theorems 1 and 2.

Let us first state a preliminary lemma that simplifies some of the expressions that involve expectations with respect to the random variable $\xi_i^{(t)}$.

**Lemma 7.** *Under the definition of our problem and the Assumptions 1-3 we have that:*

*(i)*

$$\mathbb{E}_{\xi_i^{(k)}} \left\| \frac{\sum_{i=1}^{n} \nabla F_i(z_i^{(k)}; \xi_i^{(k)})}{n} \right\|^2 = \mathbb{E}_{\xi_i^{(k)}} \left\| \frac{\sum_{i=1}^{n} \nabla F_i(z_i^{(k)}; \xi_i^{(k)}) - \nabla f_i(z_i^{(k)})}{n} \right\|^2 + \mathbb{E}_{\xi_i^{(k)}} \left\| \frac{\sum_{i=1}^{n} \nabla f_i(z_i^{(k)})}{n} \right\|^2$$

*(ii)*

$$\mathbb{E}_{\xi_i^{(k)}} \left\| \frac{\sum_{i=1}^{n} \left[\nabla F_i(z_i^{(k)}; \xi_i^{(k)}) - \nabla f_i(z_i^{(k)})\right]}{n} \right\|^2 \leq \frac{\sigma^2}{n}$$

*Proof.*

$$\mathbb{E}_{\xi_i^{(k)}} \left\| \frac{\sum_{i=1}^{n} \nabla F_i(z_i^{(k)}; \xi_i^{(k)})}{n} \right\|^2 = \mathbb{E}_{\xi_i^{(k)}} \left\| \frac{\sum_{i=1}^{n} \nabla F_i(z_i^{(k)}; \xi_i^{(k)}) - \nabla f_i(z_i^{(k)})}{n} + \frac{\sum_{i=1}^{n} \nabla f_i(z_i^{(k)})}{n} \right\|^2$$

$$= \mathbb{E}_{\xi_i^{(k)}} \left\| \frac{\sum_{i=1}^{n} \nabla F_i(z_i^{(k)}; \xi_i^{(k)}) - \nabla f_i(z_i^{(k)})}{n} \right\|^2$$

$$+ \mathbb{E}_{\xi_i^{(k)}} \left\| \frac{\sum_{i=1}^{n} \nabla f_i(z_i^{(k)})}{n} \right\|^2$$

$$+ 2 \left\langle \frac{\sum_{i=1}^{n} \mathbb{E}_{\xi_i^{(k)}} \nabla F_i(z_i^{(k)}; \xi_i^{(k)}) - \nabla f_i(z_i^{(k)})}{n} , \frac{\sum_{i=1}^{n} \nabla f_i(z_i^{(k)})}{n} \right\rangle$$

$$= \mathbb{E}_{\xi_i^{(k)}} \left\| \frac{\sum_{i=1}^{n} \nabla F_i(z_i^{(k)}; \xi_i^{(k)}) - \nabla f_i(z_i^{(k)})}{n} \right\|^2$$

$$+ \mathbb{E}_{\xi_i^{(k)}} \left\| \frac{\sum_{i=1}^{n} \nabla f_i(z_i^{(k)})}{n} \right\|^2. \tag{25}$$

where in the last equality the inner product becomes zero from the fact that $\mathbb{E}_{\xi_i^{(k)}} \nabla F_i(z_i^{(k)}; \xi_i^{(k)}) = \nabla f_i(z_i^{(k)})$.

$$
\mathbb{E}_{\xi_i^{(k)}} \left\| \frac{\sum_{i=1}^n \nabla F_i(z_i^{(k)}; \xi_i^{(k)}) - \sum_{i=1}^n \nabla f_i(z_i^{(k)})}{n} \right\|^2
$$

$$
= \frac{1}{n^2} \mathbb{E}_{\xi_i^{(k)}} \left\| \sum_{i=1}^n \left[ \nabla F_i(z_i^{(k)}; \xi_i^{(k)}) - \nabla f_i(z_i^{(k)}) \right] \right\|^2
$$

$$
= \frac{1}{n^2} \sum_{i=1}^n \mathbb{E}_{\xi_i^{(k)}} \left\| \nabla F_i(z_i^{(k)}; \xi_i^{(k)}) - \nabla f_i(z_i^{(k)}) \right\|^2
$$

$$
+ \frac{2}{n^2} \sum_{i \neq j} \left\langle \mathbb{E}_{\xi_i^{(k)}} \nabla F_i(z_i^{(k)}; \xi_i^{(k)}) - \nabla f_i(z_i^{(k)}), \mathbb{E}_{\xi_j^{(k)}} \nabla F_j(z_j^{(k)}; \xi_j^{(k)}) - \nabla f_j(z_j^{(k)}) \right\rangle
$$

$$
= \frac{1}{n^2} \sum_{i=1}^n \mathbb{E}_{\xi_i^{(k)}} \left\| \nabla F_i(z_i^{(k)}; \xi_i^{(k)}) - \nabla f_i(z_i^{(k)}) \right\|^2
$$

$$
\overset{\text{Bounded Variance}}{\leq} \frac{1}{n^2} \sum_{i=1}^n \sigma^2 = \frac{\sigma^2}{n}, \tag{26}
$$

$\square$

Before present the proof of next lemma let us define the conditional expectation $\mathbb{E}[\cdot | \mathcal{F}_k] = \mathbb{E}_{\xi_i^{(k)} \sim \mathcal{D}_i}[\cdot] = \mathbb{E}_{\xi_i^{(k)}}[\cdot]$. The expectation in this expression is with respect to the random choice $\xi_i^{(k)}$ for node $i \in [n]$ at the $k^{th}$ iteration. In other words, $\mathcal{F}_k$ denotes all the information generated by the stochastic gradient-push algorithm by time $t$, i.e., all the $x_i^{(k)}, z_i^{(k)}, w_i^{(k)}, y_i^{(k)}, \nabla F_i(z_i^{(k)}; \xi_i^{(k)})$ for $k = 1, \ldots, t$. In addition, we should highlight that the choices of random variables $\xi_i^k \sim \mathcal{D}_i$, $\xi_j^k \sim \mathcal{D}_j$ at the step $t$ of the algorithm, are independent for any two nodes $i \neq j \in [n]$. This is also true in the case that the two nodes follow the same distribution $\mathcal{D} = \mathcal{D}_i = \mathcal{D}_j$.

**Lemma 8.** *Let Assumptions 1-3 hold and let*

$$
D_1 = \frac{1}{2} - \frac{L^2}{2} \left( \frac{12\gamma^2 C^2 + 3\gamma C^2}{(1-q)^2 D_2} \right) \quad and \quad D_2 = 1 - \frac{\gamma^2 12 L^2 C^2}{(1-q)^2} - \frac{\gamma 3 L^2 C^2}{(1-q)^2}.
$$

*Here $C > 0$ and $q \in (0, 1)$ are the two non-negative constants defined in Lemma 3. Let $\{\mathbf{X}_k\}_{k=0}^\infty$ be the random sequence produced by (9) (Matrix representation of Algorithm 1). Then,*

$$
\frac{1}{K} \left( D_1 \sum_{k=0}^{K-1} \mathbb{E} \left\| \nabla f(\overline{\boldsymbol{x}}^{(k)}) \right\|^2 + \frac{1 - L\gamma}{2} \sum_{k=0}^{K-1} \mathbb{E} \left\| \frac{\nabla F(\mathbf{Z}^{(k)}) \mathbf{1}_n}{n} \right\|^2 \right)
$$

$$
\leq \frac{f(\overline{\boldsymbol{x}}^{(0)}) - f^*}{\gamma K} + \frac{L\gamma \sigma^2}{2n} + \frac{4L^2 \gamma^2 C^2 \sigma^2 + 12 L^2 \gamma^2 C^2 \zeta^2}{2(1-q)^2 D_2} + \frac{\gamma L^2 C^2 \sigma^2 + 3 L^2 \gamma C^2 \zeta^2}{2K(1-q)^2 D_2}
$$

$$
+ \left( \frac{L^2 C + 2 L^2 \gamma C^2}{2(1-q)^2 D_2 K} \right) \frac{\sum_{i=1}^n \left\| \boldsymbol{x}_i^{(0)} \right\|^2}{n}.
$$

*Proof.*

$$f\left(\overline{\boldsymbol{x}}^{(k+1)}\right) = f\left(\frac{\mathbf{X}^{(k+1)}\mathbf{1}_n}{n}\right) \qquad \overset{(9)}{=} \qquad f\left(\frac{\mathbf{X}^{(k)}[\mathbf{P}^{(k)}]^\top \mathbf{1}_n - \gamma\nabla F(\mathbf{Z}^{(k)},\xi^{(k)})[\mathbf{P}^{(k)}]^\top \mathbf{1}_n}{n}\right)$$

$$\overset{(10)}{=} \qquad f\left(\frac{\mathbf{X}^{(k)}\mathbf{1}_n}{n} - \frac{\gamma\nabla F(\mathbf{Z}^{(k)},\xi^k)\mathbf{1}_n}{n}\right)$$

$$\overset{L-smooth}{\leq} \qquad f\left(\frac{\mathbf{X}^{(k)}\mathbf{1}_n}{n}\right) - \gamma\left\langle\nabla f\left(\frac{\mathbf{X}^{(k)}\mathbf{1}_n}{n}\right), \frac{\nabla F(\mathbf{Z}^{(k)},\xi^{(k)})\mathbf{1}_n}{n}\right\rangle$$

$$+ \frac{L\gamma^2}{2}\left\|\frac{\nabla F(\mathbf{Z}^{(k)},\xi^{(k)})\mathbf{1}_n}{n}\right\|^2 \tag{27}$$

Taking expectations of both sides with respect to $\mathcal{F}_k$:

$$\mathbb{E}\left[f\left(\frac{\mathbf{X}^{(k+1)}\mathbf{1}_n}{n}\right)|\mathcal{F}_k\right] \qquad \leq \qquad f\left(\frac{\mathbf{X}^{(k)}\mathbf{1}_n}{n}\right) - \gamma\left\langle\nabla f\left(\frac{\mathbf{X}^{(k)}\mathbf{1}_n}{n}\right), \frac{\nabla F(\mathbf{Z}^{(k)})\mathbf{1}_n}{n}\right\rangle$$

$$+ \frac{L\gamma^2}{2}\mathbb{E}\left[\left\|\frac{\nabla F(\mathbf{Z}^{(k)},\xi^{(k)})\mathbf{1}_n}{n}\right\|^2|\mathcal{F}_k\right]$$

$$\overset{Lemma\ 7[i]}{=} \qquad f\left(\frac{\mathbf{X}^{(k)}\mathbf{1}_n}{n}\right) - \gamma\left\langle\nabla f\left(\frac{\mathbf{X}^{(k)}\mathbf{1}_n}{n}\right), \frac{\nabla F(\mathbf{Z}^{(k)})\mathbf{1}_n}{n}\right\rangle$$

$$+ \frac{L\gamma^2}{2}\mathbb{E}\left[\left\|\frac{\sum_{i=1}^n \nabla F_i(z_i^{(k)};\xi_i^{(k)}) - \sum_{i=1}^n \nabla f_i(z_i^{(k)})}{n}\right\|^2|\mathcal{F}_k\right]$$

$$+ \frac{L\gamma^2}{2}\mathbb{E}[\left\|\frac{\sum_{i=1}^n \nabla f_i(\boldsymbol{z}_i^{(k)})}{n}\right\|^2|\mathcal{F}_k]$$

$$\overset{Lemma\ 7[ii]}{\leq} \qquad f\left(\frac{\mathbf{X}^{(k)}\mathbf{1}_n}{n}\right) - \gamma\left\langle\nabla f\left(\frac{\mathbf{X}^{(k)}\mathbf{1}_n}{n}\right), \frac{\nabla F(\mathbf{Z}^{(k)})\mathbf{1}_n}{n}\right\rangle$$

$$+ \frac{L\gamma^2\sigma}{2n} + \frac{L\gamma^2}{2}\mathbb{E}\left[\left\|\frac{\sum_{i=1}^n \nabla f_i(\boldsymbol{z}_i^{(k)})}{n}\right\|^2|\mathcal{F}_k\right]$$

$$= \qquad f\left(\frac{\mathbf{X}^{(k)}\mathbf{1}_n}{n}\right) - \frac{\gamma}{2}\left\|\nabla f\left(\frac{\mathbf{X}^{(k)}\mathbf{1}_n}{n}\right)\right\|^2 - \frac{\gamma}{2}\left\|\frac{\nabla F(\mathbf{Z}^{(k)})\mathbf{1}_n}{n}\right\|^2,$$

$$+ \frac{\gamma}{2}\left\|\nabla f\left(\frac{\mathbf{X}^{(k)}\mathbf{1}_n}{n}\right) - \frac{\nabla F(Z^{(k)})\mathbf{1}_n}{n}\right\|^2 + \frac{L\gamma^2\sigma^2}{2n}$$

$$+ \frac{L\gamma^2}{2}\mathbb{E}\left[\left\|\frac{\sum_{i=1}^n \nabla f_i(\boldsymbol{z}_i^{(k)})}{n}\right\|^2|\mathcal{F}_k\right] \tag{28}$$

where in the last step above we simply expand the inner product.

Taking expectations again and using the tower property, we get

$$
\begin{aligned}
\mathbb{E}\left[f\left(\frac{\mathbf{X}^{(k+1)}\mathbf{1}_n}{n}\right)\right] \ \leq\ & \mathbb{E}\left[f\left(\frac{\mathbf{X}^{(k)}\mathbf{1}_n}{n}\right)\right] - \frac{\gamma}{2}\mathbb{E}\left[\left\|\nabla f\left(\frac{\mathbf{X}^{(k)}\mathbf{1}_n}{n}\right)\right\|^2\right] - \frac{\gamma}{2}\mathbb{E}\left[\left\|\frac{\nabla F(\mathbf{Z}^{(k)})\mathbf{1}_n}{n}\right\|^2\right], \\
+\ & \frac{\gamma}{2}\mathbb{E}\left[\left\|\nabla f\left(\frac{\mathbf{X}^{(k)}\mathbf{1}_n}{n}\right) - \frac{\nabla F(\mathbf{Z}^{(k)})\mathbf{1}_n}{n}\right\|^2\right] + \frac{L\gamma^2\sigma^2}{2n} \\
+\ & \frac{L\gamma^2}{2}\mathbb{E}\left[\left\|\frac{\sum_{i=1}^{n}\nabla f_i(\boldsymbol{z}_i^{(k)})}{n}\right\|^2\right] \\
=\ & \mathbb{E}\left[f\left(\frac{\mathbf{X}^{(k)}\mathbf{1}_n}{n}\right)\right] - \frac{\gamma}{2}\mathbb{E}\left[\left\|\nabla f\left(\frac{\mathbf{X}^{(k)}\mathbf{1}_n}{n}\right)\right\|^2\right] - \frac{\gamma - L\gamma^2}{2}\mathbb{E}\left[\left\|\frac{\nabla F(\mathbf{Z}^{(k)})\mathbf{1}_n}{n}\right\|^2\right], \\
+\ & \frac{\gamma}{2}\mathbb{E}\left[\left\|\nabla f\left(\frac{\mathbf{X}^{(k)}\mathbf{1}_n}{n}\right) - \frac{\nabla F(\mathbf{Z}^{(k)})\mathbf{1}_n}{n}\right\|^2\right] + \frac{L\gamma^2\sigma^2}{2n} \quad (29)
\end{aligned}
$$

Let us now focus on find an upper bound for the quantity $\mathbb{E}\left[\left\|\nabla f\left(\frac{\mathbf{X}^{(k)}\mathbf{1}_n}{n}\right) - \frac{\nabla F(Z^{(k)})\mathbf{1}_n}{n}\right\|^2\right]$.

$$
\begin{aligned}
\mathbb{E}\left[\left\|\nabla f\left(\frac{\mathbf{X}^{(k)}\mathbf{1}_n}{n}\right) - \frac{\nabla F(Z^{(k)})\mathbf{1}_n}{n}\right\|^2\right] \ =\ & \mathbb{E}\left[\left\|\nabla f\left(\frac{\sum_{i=1}^{n}\boldsymbol{x}_i^{(k)}}{n}\right) - \frac{\sum_{i=1}^{n}\nabla f_i(\boldsymbol{z}_i^{(k)})}{n}\right\|^2\right] \\
=\ & \mathbb{E}\left[\left\|\frac{1}{n}\sum_i^{n}\nabla f_i\left(\frac{\sum_{i=1}^{n}\boldsymbol{x}_i^{(k)}}{n}\right) - \frac{\sum_{i=1}^{n}\nabla f_i(\boldsymbol{z}_i^{(k)})}{n}\right\|^2\right] \\
=\ & \mathbb{E}\left[\left\|\frac{\sum_i^{n}\nabla f_i\left(\frac{\sum_{i=1}^{n}\boldsymbol{x}_i^{(k)}}{n}\right) - \sum_{i=1}^{n}\nabla f_i(\boldsymbol{z}_i^{(k)})}{n}\right\|^2\right] \\
=\ & \mathbb{E}\left[\left\|\frac{1}{n}\sum_i^{n}\left[\nabla f_i\left(\frac{\sum_{i=1}^{n}\boldsymbol{x}_i^{(k)}}{n}\right) - \nabla f_i(\boldsymbol{z}_i^{(k)})\right]\right\|^2\right] \\
\overset{Jensen}{\leq}\ & \frac{1}{n}\sum_i^{n}\mathbb{E}\left[\left\|\nabla f_i\left(\frac{\sum_{i=1}^{n}\boldsymbol{x}_i^{(k)}}{n}\right) - \nabla f_i(\boldsymbol{z}_i^{(k)})\right\|^2\right] \\
\overset{L-smooth}{\leq}\ & \frac{L^2}{n}\sum_i^{n}\mathbb{E}\left[\left\|\frac{\sum_{i=1}^{n}\boldsymbol{x}_i^{(k)}}{n} - \boldsymbol{z}_i^{(k)}\right\|^2\right] \\
=\ & \frac{L^2}{n}\sum_{i=1}^{n}Q_i^{(k)} \quad (30)
\end{aligned}
$$

Thus we have that:

$$
\begin{aligned}
\mathbb{E}\left[f\left(\frac{\mathbf{X}^{(k+1)}\mathbf{1}_n}{n}\right)\right] \ \leq\ & \mathbb{E}\left[f\left(\frac{\mathbf{X}^{(k)}\mathbf{1}_n}{n}\right)\right] - \frac{\gamma}{2}\mathbb{E}\left[\left\|\nabla f\left(\frac{\mathbf{X}^{(k)}\mathbf{1}_n}{n}\right)\right\|^2\right] - \frac{\gamma - L\gamma^2}{2}\mathbb{E}\left[\left\|\frac{\nabla F(\mathbf{Z}^{(k)})\mathbf{1}_n}{n}\right\|^2\right], \\
+\ & \frac{\gamma L^2}{2n}\sum_{i=1}^{n}Q_i^{(k)} + \frac{L\gamma^2\sigma^2}{2n} \quad (31)
\end{aligned}
$$

By rearranging:

$$\frac{\gamma}{2}\mathbb{E}[\left\|\nabla f\left(\frac{\mathbf{X}^{(k)}\mathbf{1}_n}{n}\right)\right\|^2] + \frac{\gamma - L\gamma^2}{2}\mathbb{E}[\left\|\frac{\nabla F(\mathbf{Z}^{(k)})\mathbf{1}_n}{n}\right\|^2] \leq \mathbb{E}[f\left(\frac{\mathbf{X}^{(k)}\mathbf{1}_n}{n}\right)] - \mathbb{E}[f\left(\frac{\mathbf{X}^{(k+1)}\mathbf{1}_n}{n}\right)]$$
$$+ \frac{L\gamma^2\sigma^2}{2n} + \frac{\gamma L^2}{2n}\sum_{i=1}^n Q_i^{(k)} \qquad (32)$$

Let us now sum from $k = 0$ to $k = K - 1$:

$$\frac{\gamma}{2}\sum_{k=0}^{K-1}\mathbb{E}[\left\|\nabla f\left(\frac{\mathbf{X}^{(k)}\mathbf{1}_n}{n}\right)\right\|^2] + \frac{\gamma - L\gamma^2}{2}\sum_{k=0}^{K-1}\mathbb{E}[\left\|\frac{\nabla F(\mathbf{Z}^{(k)})\mathbf{1}_n}{n}\right\|^2] \leq \sum_{k=0}^{K-1}\left[\mathbb{E}[f\left(\frac{\mathbf{X}^{(k)}\mathbf{1}_n}{n}\right)] - \mathbb{E}[f\left(\frac{\mathbf{X}^{(k+1)}\mathbf{1}_n}{n}\right)]\right]$$
$$+ \sum_{k=0}^{K-1}\frac{L\gamma^2\sigma^2}{2n} + \frac{\gamma L^2}{2n}\sum_{k=0}^{K-1}\sum_{i=1}^n \mathbb{E}[Q_i^{(k)}]$$
$$\leq \mathbb{E}[f\left(\frac{\mathbf{X}^{(0)}\mathbf{1}_n}{n}\right)] - \mathbb{E}[f\left(\frac{\mathbf{X}^{(k)}\mathbf{1}_n}{n}\right)]$$
$$+ \frac{LK\gamma^2\sigma^2}{2n} + \frac{\gamma L^2}{2}\sum_{k=0}^{K-1}\frac{1}{n}\sum_{i=1}^n Q_i^{(k)}$$
$$\leq f(\overline{\boldsymbol{x}}^{(0)}) - f^*$$
$$+ \underbrace{\frac{LK\gamma^2\sigma^2}{2n} + \frac{\gamma L^2}{2}\sum_{k=0}^{K-1}\frac{1}{n}\sum_{i=1}^n Q_i^{(k)}}_{M_k} \qquad (33)$$

For the last inequality above, recall that with $f^*$ we define the optimal solution of our problem.

Using the bound for the expression $\sum_{k=0}^{K-1} M_k$ from Lemma 6 we obtain:

$$\frac{\gamma}{2}\sum_{k=0}^{K-1}\mathbb{E}[\left\|\nabla f\left(\frac{\mathbf{X}^{(k)}\mathbf{1}_n}{n}\right)\right\|^2] + \frac{\gamma - L\gamma^2}{2}\sum_{k=0}^{K-1}\mathbb{E}[\left\|\frac{\nabla F(\mathbf{Z}^{(k)})\mathbf{1}_n}{n}\right\|^2]$$
$$\leq f(\overline{\boldsymbol{x}}^{(0)}) - f^* + \frac{LK\gamma^2\sigma^2}{2n}$$
$$+ \frac{\gamma L^2}{2}\frac{4\gamma^2 C^2\sigma^2 K + \gamma C^2\sigma^2}{(1-q)^2 D_2} + \frac{\gamma L^2}{2}\frac{12\gamma^2 C^2\zeta^2 K + 3\gamma C^2\zeta^2}{(1-q)^2 D_2}$$
$$+ \frac{\gamma L^2}{2}\left(\frac{12\gamma^2 C^2 + 3\gamma C^2}{(1-q)^2 D_2}\right)\sum_{k=0}^K \mathbb{E}\left\|\nabla f(\overline{\boldsymbol{x}}^{(k)})\right\|^2$$
$$+ \frac{\gamma L^2}{2}\left(\frac{C^2 + 2\gamma C^2}{(1-q)^2 D_2}\right)\frac{\sum_{i=1}^n \left\|\boldsymbol{x}_i^{(0)}\right\|^2}{n}.$$

By rearranging and dividing all terms by $\gamma K$ we obtain:

$$\frac{1}{K}\left(\left[\frac{1}{2} - \frac{L^2}{2}\left(\frac{12\gamma^2 C^2 + 3\gamma C^2}{(1-q)^2 D_2}\right)\right]\sum_{k=0}^{K-1}\mathbb{E}\left\|\nabla f(\overline{\boldsymbol{x}}^{(k)})\right\|^2 + \frac{1 - L\gamma}{2}\sum_{k=0}^{K-1}\mathbb{E}\left\|\frac{\nabla F(\mathbf{Z}^{(k)})\mathbf{1}_n}{n}\right\|^2\right)$$
$$\leq \frac{f(\overline{\boldsymbol{x}}^{(0)}) - f^*}{\gamma K} + \frac{L\gamma\sigma^2}{2n} + \frac{4L^2\gamma^2 C^2\sigma^2 + 12L^2\gamma^2 C^2\zeta^2}{2(1-q)^2 D_2} + \frac{\gamma L^2 C^2\sigma^2 + 3L^2\gamma C^2\zeta^2}{2K(1-q)^2 D_2}$$
$$+ \left(\frac{L^2 C^2 + 2L^2\gamma C^2}{2(1-q)^2 D_2 K}\right)\frac{\sum_{i=1}^n \left\|\boldsymbol{x}_i^{(0)}\right\|^2}{n}.$$

By defining $D_1 = \left[\frac{1}{2} - \frac{L^2}{2}\left(\frac{12\gamma^2 C^2 + 3\gamma C^2}{(1-q)^2 D_2}\right)\right]$ the proof is complete. $\qquad\square$

### C.3 Proofs of Main Theorems

Having present all of the above Lemmas we are now ready to provide the proofs of main Theorems 1 and 2.

#### C.3.1 Proof of Theorem 1

Let $\gamma \leq \min \left\{ \dfrac{(1-q)^2}{60L^2C^2}, 1 \right\}$. Then:

$$D_2 = 1 - \frac{\gamma^2 12 L^2 C^2}{(1-q)^2} - \frac{\gamma 3 L^2 C^2}{(1-q)^2} \overset{(\gamma^2 < \gamma)}{\geq} 1 - \frac{\gamma 15 L^2 C^2}{(1-q)^2} \geq 1 - \frac{1}{4} \geq \frac{1}{2}$$

and

$$D_1 = \frac{1}{2} - \frac{L^2}{2} \left( \frac{12\gamma^2 C^2 + 3\gamma C^2}{(1-q)^2 D_2} \right) \overset{(\gamma^2 < \gamma)}{\geq} \frac{1}{2} - \frac{15\gamma C^2 L^2}{2(1-q)^2 D_2} \geq \frac{1}{2} - \frac{1}{8D_2} \geq \frac{1}{4}$$

By substituting the above bounds into the result of Lemma 8 and by removing the second term of left hand side we obtain:

$$
\begin{aligned}
\frac{1}{4} \frac{\sum_{k=0}^{K-1} \mathbb{E}\left\|\nabla f(\overline{\boldsymbol{x}}^{(k)})\right\|^2}{K} &= \frac{1}{K}\left( \frac{1}{4} \sum_{k=0}^{K-1} \mathbb{E}\left\|\nabla f(\overline{\boldsymbol{x}}^{(k)})\right\|^2 + \frac{1 - L\gamma}{2} \sum_{k=0}^{K-1} \mathbb{E}\left\|\frac{\nabla F(\mathbf{Z}_k)\mathbf{1}_n}{n}\right\|^2 \right) \\
&\leq \frac{f(\overline{\boldsymbol{x}}^{(0)}) - f^*}{\gamma K} + \frac{L\gamma\sigma^2}{2n} + \frac{4L^2\gamma^2 C^2\sigma^2 + 12L^2\gamma^2 C^2\zeta^2}{(1-q)^2} + \frac{\gamma L^2 C^2\sigma^2 + 3L^2\gamma C^2\zeta^2}{K(1-q)^2} \\
&\quad + \left( \frac{L^2 C + 2L^2\gamma C^2}{(1-q)^2 K} \right) \frac{\sum_{i=1}^{n}\left\|\boldsymbol{x}_i^{(0)}\right\|^2}{n}
\end{aligned}
\tag{34}
$$

Let us now substitute in the above expression $\gamma = \sqrt{\frac{n}{K}}$. This can be done due to the lower bound (see equation 6) on the total number of iterations $K$ where guarantees that $\sqrt{\frac{n}{K}} \leq \min\left\{ \dfrac{(1-q)^2}{60L^2C^2}, 1 \right\}$.

$$
\begin{aligned}
\frac{1}{4} \frac{\sum_{k=0}^{K-1} \mathbb{E}\left\|\nabla f(\overline{\boldsymbol{x}}^{(k)})\right\|^2}{K} &\leq \frac{f(\overline{\boldsymbol{x}}^{(0)}) - f^*}{\gamma K} + \frac{L\gamma\sigma^2}{2n} + \gamma^2 \frac{4L^2 C^2\sigma^2 + 12L^2 C^2\zeta^2}{(1-q)^2} + \gamma \frac{L^2 C^2\sigma^2 + 3L^2 C^2\zeta^2}{K(1-q)^2} \\
&\quad + \frac{L^2 C}{(1-q)^2 K} \frac{\sum_{i=1}^{n}\left\|\boldsymbol{x}_i^{(0)}\right\|^2}{n} + \gamma \frac{2L^2 C^2}{(1-q)^2 K} \frac{\sum_{i=1}^{n}\left\|\boldsymbol{x}_i^{(0)}\right\|^2}{n} \\
&\overset{\gamma=\sqrt{\frac{n}{K}}}{=} \frac{f(\overline{\boldsymbol{x}}^{(0)}) - f^*}{\sqrt{nK}} + \frac{L\sigma^2}{2\sqrt{nK}} + \frac{n}{K} \frac{4L^2 C^2\sigma^2 + 12L^2 C^2\zeta^2}{(1-q)^2} + \sqrt{\frac{n}{K}} \frac{L^2 C^2\sigma^2 + 3L^2 C^2\zeta^2}{K(1-q)^2} \\
&\quad + \frac{L^2 C^2}{(1-q)^2 K} \frac{\sum_{i=1}^{n}\left\|\boldsymbol{x}_i^{(0)}\right\|^2}{n} + \sqrt{\frac{n}{K}} \frac{2L^2 C^2}{(1-q)^2 K} \frac{\sum_{i=1}^{n}\left\|\boldsymbol{x}_i^{(0)}\right\|^2}{n} \\
&= \frac{f(\overline{\boldsymbol{x}}^{(0)}) - f^* + \frac{L}{2}\sigma^2}{\sqrt{nK}} + \frac{L^2 C^2}{K(1-q)^2}\left[ (4\sigma^2 + 12\zeta^2)n + \frac{\sum_{i=1}^{n}\left\|\boldsymbol{x}_i^{(0)}\right\|^2}{n} \right] \\
&\quad + \frac{\sqrt{n}L^2 C^2}{\sqrt{K}(1-q)^2 K}\left[ \sigma^2 + 3L^2 C^2\zeta^2 + 2\frac{\sum_{i=1}^{n}\left\|\boldsymbol{x}_i^{(0)}\right\|^2}{n} \right]
\end{aligned}
\tag{35}
$$

Using again the assumption on the lower bound (6) of the total number of iterations $K$, the last two terms of the above expression are bounded by the first term. Thus,

$$\frac{1}{4} \frac{\sum_{k=0}^{K-1} \mathbb{E}\left\|\nabla f(\overline{\boldsymbol{x}}^{(k)})\right\|^2}{K} \leq 3\frac{f(\overline{\boldsymbol{x}}^{(0)}) - f^* + \frac{L}{2}\sigma^2}{\sqrt{nK}} \tag{36}$$

### C.3.2 PROOF OF THEOREM 2

*Proof.* From Lemma 6 we have that:

$$
\begin{aligned}
\frac{1}{K} \sum_{k=0}^{K-1} M^{(k)} \leq & \left( \gamma^2 \frac{4C^2}{(1-q)^2 D_2} \right) \sigma^2 + \left( \gamma \frac{C^2}{(1-q)^2 D_2} \right) \frac{\sigma^2}{K} \\
& + \left( \gamma^2 \frac{12C^2}{(1-q)^2 D_2} \right) \zeta^2 + \left( \frac{\gamma 3 C^2}{(1-q)^2 D_2} \right) \frac{\zeta^2}{K} \\
& + \left( \frac{C^2}{(1-q)^2 D_2 K} + \gamma \frac{2C^2}{(1-q)^2 D_2 K} \right) \frac{\sum_{i=1}^{n} \left\| \boldsymbol{x}_i^{(0)} \right\|^2}{n} \\
& + \left( \gamma^2 \frac{12C^2}{(1-q)^2 D_2} + \gamma \frac{3C^2}{(1-q)^2 D_2} \right) \frac{\sum_{k=0}^{K-1} \mathbb{E} \left\| \nabla f(\overline{\boldsymbol{x}}^{(k)}) \right\|^2}{K} \quad (37)
\end{aligned}
$$

Using the assumptions of Theorem 1 and stepsize $\gamma = \sqrt{\frac{n}{K}}$:

$$
\begin{aligned}
\frac{1}{K} \sum_{k=0}^{K-1} M^{(k)} \leq & \left( \frac{n}{K} \frac{4C^2}{(1-q)^2 D_2} \right) \sigma^2 + \left( \sqrt{\frac{n}{K}} \frac{C^2}{(1-q)^2 D_2} \right) \frac{\sigma^2}{K} \\
& + \left( \frac{n}{K} \frac{12C^2}{(1-q)^2 D_2} \right) \zeta^2 + \left( \frac{\sqrt{\frac{n}{K}} 3C^2}{(1-q)^2 D_2} \right) \frac{\zeta^2}{K} \\
& + \left( \frac{C^2}{(1-q)^2 D_2 K} + \sqrt{\frac{n}{K}} \frac{2C^2}{(1-q)^2 D_2 K} \right) \frac{\sum_{i=1}^{n} \left\| \boldsymbol{x}_i^{(0)} \right\|^2}{n} \\
& + \left( \frac{n}{K} \frac{12C^2}{(1-q)^2 D_2} + \sqrt{\frac{n}{K}} \frac{3C^2}{(1-q)^2 D_2} \right) \frac{12 \left[ f(\overline{\boldsymbol{x}}^{(0)}) - f^* + \frac{L}{2} \sigma^2 \right]}{\sqrt{nK}} \\
= & \frac{1}{K} \left[ \frac{4nC^2\sigma^2}{(1-q)^2 D_2} + \frac{12nC^2\zeta^2}{(1-q)^2 D_2} + \frac{C^2 \sum_{i=1}^{n} \left\| \boldsymbol{x}_i^{(0)} \right\|^2}{n(1-q)^2 D_2} + \frac{3\sqrt{n}C^2 12 \left[ f(\overline{\boldsymbol{x}}^{(0)}) - f^* + \frac{L}{2} \sigma^2 \right]}{\sqrt{n}(1-q)^2 D_2} \right] \\
+ & \frac{1}{K\sqrt{K}} \left[ \frac{n\sigma^2 C^2}{(1-q)^2 D_2} + \frac{\frac{n}{3} C^2 \zeta^2}{(1-q)^2 D_2} + \frac{2C^2 \sum_{i=1}^{n} \left\| \boldsymbol{x}_i^{(0)} \right\|^2}{(1-q)^2 D_2 \sqrt{n}} + \frac{144\sqrt{n}C^2 \left[ f(\overline{\boldsymbol{x}}^{(0)}) - f^* + \frac{L}{2} \sigma^2 \right]}{(1-q)^2 D_2} \right] \\
= & O\left( \frac{1}{K} + \frac{1}{K\sqrt{K}} \right) \quad (38)
\end{aligned}
$$

where the Big O notation swallows all constants of our setting $\left( n, L, \sigma, \zeta, C, q, \sum_{i=1}^{n} \left\| x_i^{(0)} \right\|^2 \text{ and } f(\overline{\boldsymbol{x}}^{(0)}) - f^* \right)$.

Now using the above upper bound equation 38 and result of Theorem 1 we obtain:

$$
\begin{aligned}
\frac{1}{K}\sum_{k=0}^{K-1}\frac{1}{n}\sum_{i=1}^{n}\mathbb{E}\left\|\nabla f(\boldsymbol{z}_i^k)\right\|^2 \quad & = \quad \frac{1}{K}\sum_{k=0}^{K-1}\frac{1}{n}\sum_{i=1}^{n}\mathbb{E}\left\|\nabla f(\boldsymbol{z}_i^k)+\nabla f(\overline{\boldsymbol{x}}^{(k)})-\nabla f(\overline{\boldsymbol{x}}^{(k)})\right\|^2 \\[2mm]
& \leq \quad \frac{1}{K}\sum_{k=0}^{K-1}\frac{1}{n}\sum_{i=1}^{n}2\mathbb{E}\left\|\nabla f(\boldsymbol{z}_i^k)-\nabla f(\overline{\boldsymbol{x}}^{(k)})\right\|^2+2\mathbb{E}\left\|\nabla f(\overline{\boldsymbol{x}}^{(k)})\right\|^2 \\[2mm]
& = \quad \frac{1}{K}\sum_{k=0}^{K-1}\frac{1}{n}\sum_{i=1}^{n}2\mathbb{E}\left\|\nabla f(\boldsymbol{z}_i^k)-\nabla f(\overline{\boldsymbol{x}}^{(k)})\right\|^2+\frac{1}{K}\sum_{k=0}^{K-1}\frac{1}{n}\sum_{i=1}^{n}2\mathbb{E}\left\|\nabla f(\overline{\boldsymbol{x}}^{(k)})\right\|^2 \\[2mm]
& = \quad 2\frac{1}{K}\sum_{k=0}^{K-1}\frac{1}{n}\sum_{i=1}^{n}\mathbb{E}\left\|\nabla f(\boldsymbol{z}_i^k)-\nabla f(\overline{\boldsymbol{x}}^{(k)})\right\|^2+2\frac{1}{K}\sum_{k=0}^{K-1}\mathbb{E}\left\|\nabla f(\overline{\boldsymbol{x}}^{(k)})\right\|^2 \\[2mm]
& \overset{L-smooth}{=} \quad 2L^2\frac{1}{K}\sum_{k=0}^{K-1}\frac{1}{n}\sum_{i=1}^{n}\mathbb{E}\left\|\boldsymbol{z}_i^k-\overline{\boldsymbol{x}}^{(k)}\right\|^2+2\frac{1}{K}\sum_{k=0}^{K-1}\mathbb{E}\left\|\nabla f(\overline{\boldsymbol{x}}^{(k)})\right\|^2 \\[2mm]
& \overset{(38)+(36)}{\leq} \quad O\left(\frac{1}{\sqrt{nK}}+\frac{1}{K}+\frac{1}{K^{3/2}}\right)
\end{aligned}
\tag{39}
$$

where again the Big O notation swallows all constants of our setting $\left(n,L,\sigma,\zeta,C,q,\sum_{i=1}^{n}\left\|x_i^{(0)}\right\|^2 \text{ and} f(\overline{\boldsymbol{x}}^{(0)})-f^*\right)$. □

