# OpenReview forum: "Stochastic Gradient Push for Distributed Deep Learning"
_ICLR.cc/2019/Conference_

### Official Review · AnonReviewer1 · 2018-11-02
**Interesting application of PushSum, but how does it fare relative to AD-PSGD?**

**Rating:** 6
**Confidence:** 3

**Review:**

# overview
This paper leverages a consensus based approach for computing and communicating approximate gradient averages to each node running a decentralized version of stochastic gradient descent.

Though the PushSum idea isn't new, its application to distributed SGD and corresponding convergence analysis represents a valuable contribution, and the experimental results indicate a potentially large speedup (in highly variable or latent networks) without substantially sacrificing model accuracy.

The paper itself is reasonably comprehensive but does miss out on comparisons with more recent but equally promising approaches, namely AD-PSGD.

# pros
* Empirically shown to be significantly faster than SGD, D-PSGD in high-latency, communication bound configurations which is a fairly common real-world setup. There is an accuracy tradeoff at work here, but performance doesn't seem to suffer too much as the node count scales.
* introduces and proves theoretical guarantees for SGP approximate distributed average convergence for smooth, non-convex case, including upper bounded convergence rates.

# cons
* biggest criticism is that AD-PSGD from Lian et al 2018 is not included in experimental comparisons even though the paper is referenced. Authors state that asynchronous methods typically generalize worse than their synchonous counterparts but that isn't what Lian et al found in their comparison with D-PSGD (see table 2 and 3 from their paper). This comparison would be particularly interesting as AD-PSGD also performs well in the high network latency regime that SGP is touted for.
* would've liked to see comparison on other tasks beyond just image classification on ResNet.

# other comments
* Didn't see mention of specific iteration count value(s) K used in experiments or hyperparameters A.3. Since it bounds the convergence rate, this would be important to include.
* Found a few small typos:
  * pg. 5: Relatively -> Relative
  * pg. 7, fig. 2: part -> par
  * pg. 8, sec. 5.3. par. 2: achieves -> achieved
  * pg. 8, sec. 5.3, par. 2: "neighbors also to increases" (drop "to")
  * pg. 12, sec. A.2: "send-buffer to filled" -> "send-buffer to be filled"

---

> ### Author Response · Authors · 2018-11-15
> **Thanks for your suggestions**
>
> Thank you for your feedback and for finding our work to be a valuable contribution.
>
> *About the comparison with AD-PSGD:*
> Thank you for this suggestion! In order to provide a comparison with AD-PSGD, we have re-implemented it since there is no openly-available reference implementation. We discuss the results here, and we are also preparing an updated version of the paper that will include these new results.
>
> We compared AD-PSGD to AllReduce SGD, D-PSGD and SGP on 32 GPUs in a low-bandwidth/high-latency scenario (10Gbps Ethernet). Our AD-PSGD implementation achieves 75.48% validation accuracy, which is better than what Lian et al. (2018) report in a similar setting (74.66%). However, AD-PSGD is outperformed, in terms of validation accuracy, by both AllReduce SGD and SGP, which respectively achieve 76.23% and 76.33%.
>
> Observe that our implementation of AllReduce SGD achieves better test accuracy than the results reported in Table 3 of Lian et al. (2018) for both AllReduce SGD and AD-PSGD for different network sizes (32, 64 and 128GPUs). SGP also achieves better test accuracy than the values reported in Lian et al. (2018).
>
> In terms of computational time, our implementation of AD-PSGD is faster than both AllReduce SGD and D-PSGD with an average time-per-iteration of 0.48ms, but it is not faster than SGP, which has an average time-per-iteration of 0.38ms. Note that it is difficult to directly compare times between our results and those reported in Lian et al. (2018) since the experiments were conducted on different systems and with different implementations. Recall that in AD-PSGD the nodes are partitioned into two equal-sized sets: active and passive nodes. The passive nodes in AD-PSGD do not initiate communication, and we observe that they run faster (in iterations per second) than the average node in SGP. On the other hand, active nodes (which do initiate communication) initiate one push-pull communication (i.e., send and receive) with a passive node at every update to have doubly stochastic mixing matrices, and we observe that the active nodes run substantially slower than SGP nodes.
>
> Finally, we would like to emphasize that the contributions of AD-PSGD and SGP can be seen as being orthogonal. By combining the two (leverage the PushSum gossip protocol in an asynchronous manner), one could expect to further speed up SGP. We leave this as a promising line of investigation for future work.
>
> We plan to release our implementations of SGP and D-PSGD and AD-PSGD to foster more research in this area and ease evaluation and comparison between algorithms. Each algorithm is implemented as a Pytorch module in a similar manner as torch.distributed.DistributedDataParallel.
>
> *About comparison on other tasks:*
> The ImageNet dataset is a standard relatively large-scale dataset that has been used in many papers looking at training deep neural networks in a distributed setting (Goyal et al, 2017; Lian et al., 2017; Lian et al., 2018). For this reason we chose to focus the empirical evaluation on this datasets. We agree that a comparison with other datasets would be valuable and we intend to investigate this in a near future.
>
> *About the #other comments:*
> Thanks again for the suggestions! We will include the global iteration count 'K'. In particular, as we scale up the number of nodes 'n', we scale down the number of iterations to 'K/n'. That is, 32-node runs involve 8x fewer global iterations than 4-node runs.
>
> We are incorporating the information from our responses into a revised version of our paper.

---

> ### Author Response · Authors · 2018-12-10
> **Follow-up**
>
> Thank you again for your comments and suggestions. Have our responses and the changes we made to the manuscript addressed all of your concerns?

---

> > ### Comment · AnonReviewer1 · 2018-12-17
> > **thanks for adding AD-PSGD**
> >
> > Thank you for taking the time and effort to address my biggest concern by implementing and comparing AD-PSGD relative to SGP and others.
> >
> > Looking at your updated manuscript I don't see mention of the active/passive time differences you note in the comment below, and the timing reported in Table 2 indicates that the AD-PSGD time is actually faster than SGP across each of the node counts measured (not slower as you mention in your comment).  As you point out, AD-PSGD isn't generally as accurate as SGP, and indeed combining these orthogonal approaches does make for an interesting line of future work.  Note that AD-PSGD is missing from the 3 plots in Figure 1 and should be included there for completeness/clarity.
> >
> > I also don't see any updates in the manuscript around the global iteration count value 'K' used in the experiments (would've expected to see it mentioned in A.3).

---

> > > ### Author Response · Authors · 2018-12-19
> > > **thanks for your reply, with clarifications**
> > >
> > > >> Looking at your updated manuscript I don't see mention of the active/passive time differences you note in the comment below, and the timing reported in Table 2 indicates that the AD-PSGD time is actually faster than SGP across each of the node counts measured (not slower as you mention in your comment).
> > >
> > > Between the time when we posted our original response to you and when we updated the paper, we improved our implementation of AD-PSGD. We mentioned this in the post “Revised Version 1.0” above (repeated here for convenience):
> > > “ * We added a comparison with the asynchronous AD-PSGD method on the high-latency interconnect in section 5.1.
> > > * We find that AD-PSGD exhibits virtually no accuracy degradation relative to D-PSGD over the range from 32—256GPUs, and provides a substantial speedup in the high latency case, bolstering the findings in Lian et al. 2018
> > > * We find that AD-PSGD runs slightly faster than SGP at the expense of slightly lower training/validation accuracy (except in the 256GPU case where the accuracies are very similar; cf. Tables 1 and 2 in Section 5.1).
> > > * note: in our original reply we had implemented AD-PSGD exactly as described in Appendix A of Lian et al. 2018. However, we noticed that AD-PSGD was bottlenecked by the Python GIL in this multithreaded case, so we implemented a completely concurrent multi-processed version of AD-PSGD in PyTorch. The algorithm is implemented as a PyTorch module in a similar manner as torch.distributed.DistributedDataParallel, and, as stated previously, we plan to release our implementations to foster more research in this area. Of course it is difficult to compare the runtime of implementations on different systems, but just for perspective, our new implementation of AD-PSGD runs at a rate of ~226s/epoch and achieves an accuracy of ~76% in the 64GPU case, compared to the rate of 264s/epoch and accuracy of ~74% presented in Lian et al. 2018 for a similar 10Gbps network. This corresponds to a 1 hour reduction in training time with a 2% improvement in the top-1 validation accuracy.
> > > * Again we emphasize that this asynchronous line of work is orthogonal, and that by combining the two approaches (leveraging the PushSum protocol in an asynchronous manner), one can expect to further speed up SGP. We leave this as a promising line of investigation for future work.
> > > * We updated the description of AD-PSGD in related work section to better reflect the findings of Lian et al. 2018.
> > > * We reported the number of iterations K used by the different training runs in appendix B.1”
> > >
> > > The average time per iteration reported in the table is obtained by dividing the time to complete a total number of iterations equivalent to 90 epochs across the entire network, divided by corresponding total number of iterations.
> > >
> > > After making this improvement in our implementation, AD-PSGD is indeed always faster than SGP. The wording in the revised paper reflects this too.
> > >
> > > Unfortunately, at this point (since rebuttal period closed on Nov 26) we are unable to post an updated paper on OpenReview. We will add a discussion about the difference in timing between active and passive nodes, as requested, once the revisions feature becomes unlocked again. Please note, though, that in the revised paper, the caption of Table 2 does say “The average time per iteration for the asynchronous method is calculated by dividing the average time per epoch by the total number of iterations per epoch."
> > >
> > >
> > > >> Note that AD-PSGD is missing from the 3 plots in Figure 1 and should be included there for completeness/clarity.
> > > >>
> > > >> I also don't see any updates in the manuscript around the global iteration count value 'K' used in the experiments (would've expected to see it mentioned in A.3).
> > >
> > >
> > > We will also update the three plots in Figure 1 as requested once the revisions feature becomes unlocked again. The total number of iterations for each experiment (depending on the number of nodes) is currently shown Table 3 in Appendix B.1. We will also move this to Appendix A.3, as requested.

---

### Official Review · AnonReviewer2 · 2018-11-03
**An interesting work that needs some clarification**

**Rating:** 6
**Confidence:** 4

**Review:**

This paper demonstrates the benefit of stochastic gradient push (SGP) in the distributed training of neural networks. The contributions are twofold: (1) the paper proves the convergence of SGP for nonconvex smooth functions and gives a reasonable estimation of the convergence rate; (2) the paper did many experiments and shows the SGP can achieve a significant speed-up in the low-latency environment without sacrificing too much predictive performance.

I like this work. Although SGP is not the contribution of this paper, the paper strengthens the algorithm in theoretical perspective and broadens its usage into deep neural network training.

One thing the authors need to clarify is how to generate/choose P^{(k)}. This is different from Markov-Chain, since time invariant MCs will fix the transition kernels. Here P^{(k)} seems to be randomly sampled for each k. According to the theory, P^{(k)} also must correspond to a strongly connected graph. Then it is better to explain how to control the sparsity of each P^{(k)} and sample its values. And if P^{(k)} needs to vary each step, how to notify P^{(k)} to all the nodes in the cluster and how to maintain its consistency across the nodes? This seems another communication workload, but the paper never mentions that.

---

> ### Author Response · Authors · 2018-11-15
> **Thank you for liking our work (Response Part 1)**
>
> Thank you for your feedback and for liking our work.
>
> *About the choice of mixing matrices P^{(k)}:*
> In our experiments we choose the mixing matrices P^{(k)} in a deterministic manner, by cycling through neighbors in the directed exponential graph described in Appendix A. For example, when n=32, each node has 5 neighbors. For i in {0,1,...,n-1}, node i has neighbors (i + 2j) mod n for j in {0,1,2,3,4}. In our default implementation, at every iteration, each node sends to one neighbor, so every column of P^{(k)} has two non-zero entries, both of which are equal to 0.5. The diagonal entries are always equal to 0.5, and P_{i',i}^{(k)} = 0.5 where i' = (i + 2**j) mod n and j = k mod 5.
>
> We made this decision because this sequence of matrices has very nice mixing properties. To understand this choice, let's focus on simply averaging using linear iterations x(k+1) = P(k) x(k), ignoring the additional gradient term being injected at each node for optimization. (We're using Push-Sum for averaging, so this is just focusing on how well we do averaging while ignoring additional complications for doing optimization simultaneously.)
>
> Since x(k) = P(k-1) P(k-2) ... P(1) P(0) x(0), the worst-case averaging error after k steps is related to the second largest singular value of the product of matrices P(k-1) P(k-2) ... P(1) P(0); see, e.g., the reference Nedic et al. (2018) cited in our paper for a detailed derivation. Note that, since the matrices are column stochastic, the second largest singular value lies in the interval [0,1].
>
> The directed exponential graph has an expander-like quality: from any starting vertex, one can get to any destination after at most log2(n) steps because one can always take a step which reduces the distance to the destination by half. This also translates to good mixing properties. For a 32-node directed exponential graph, each node has 5 neighbors. After every product of 5 consecutive matrices P(4) P(3) P(2) P(1) P(0), where matrix P(k) corresponds to the step where every node i sends a message to node (i + 2**k) mod n, the second largest singular value of the product is at machine precision (2e-17); i.e., if nodes were only averaging, then they would effectively have the exact average.
>
> We could have had each node randomly sample a neighbor at every iteration, but this would have resulted in slower mixing. For example, if each node sent a message to one randomly sampled neighbor in the directed exponential graph, then after the same number of steps the (empirical average) second largest singular value of the product is around 0.4, meaning that in the worst case, the difference || x(k) - avg(0) || <= 0.4 ||x(0) - avg(0)||, where avg(0) is the average of the initial values at all nodes. Even if nodes randomly sample a neighbor uniformly from all other nodes in the network, the (expected) second largest singular value is still around 0.2 after 5 steps. Hence, random sampling would lead to nodes being much less well-synchronized. In the context of stochastic optimization, this can be interpreted as having additional noise in the step direction which, we believe, would lead to even worse accuracy. Moreover, implementation-wise, if nodes randomly and independently sample an out neighbor at every iteration, it is likely that some nodes will receive more than one message in some iterations. On the other hand, with the scheme adopted in the paper every node sends and receives exactly one message at every iteration, so the communication load is always balanced across nodes.
>
> *Regarding “... P^{(k)} also must correspond to a strongly connected graph”:*
> The theory in our paper does not require that each individual P^{(k)} correspond to a strongly connected graph. Rather (see assumption 3, specifically), if we take the union over edge sets of the graphs corresponding to B consecutive P^{(k)}'s, we need there to be a B such that this union graph is strongly connected. This is a weaker condition than requiring each graph to be strongly connected. For the specific example where the P^{(k)}'s are chosen as described above, cycling through neighbors in the directed exponential graph, when n=32 then for B=5 we have the property we need, since after 5 steps we have sent to all neighbors in the directed exponential graph, which itself is strongly connected.

---

> > ### Author Response · Authors · 2018-11-15
> > **(Response Part 2)**
> >
> >
> > *About the implementation of this scheme:*
> > The number of neighbors a node chooses to communicate with directly controls the sparsity of the mixing matrices.
> > In general, we have each node choose one neighbor from its out-neighbor set to communicate with at each iteration, cycling through the neighbors in a deterministic order. We also conducted other experiments, with less sparse mixing matrices, where nodes deterministically choose two neighbors from their out-neighbour set to communicate with at each iteration.
> >
> > Nodes do not need to know the global structure of the mixing matrix at each time step, they only need to decide which neighbor(s) they will send a message to, and with what mixing weight. Thus, there is no overhead required to coordinate this over the network. In our experiments, nodes determine their out-neighbor(s) in each iteration by deterministically cycling through their out-neighbour set. Also note that SGP is synchronous, so nodes cycle through their neighborhoods at the same rate. Fixing the order of the out-neighbour set at initialization predetermines the graph topology at each iteration.
> >
> > Thank you again for your feedback. We are preparing a revised version of the paper in which we will clarify all of these points.

---

> ### Author Response · Authors · 2018-12-10
> **Follow-up**
>
> Thank you again for your comments and suggestions. Have our responses and the changes we made to the manuscript addressed all of your concerns?

---

### Official Review · AnonReviewer3 · 2018-11-06
**Good balance of theory and practice, lackluster experiment results**

**Rating:** 6
**Confidence:** 3

**Review:**

Authors propose using gossip algorithms as a general method of computing approximate average over a set of workers approximately. Gossip algorithm approach is to perform linear iterations to compute consensus, they adapt this to practical setting by sending only to 1 or 2 neighbors at a time, and rotating the neighbors.

Experiments are reasonably comprehensive -- they compare against AllReduce on ImageNet which is a well-tuned implementation, and D-PSGD.

Their algorithm seems to trade-off latency for accuracy -- for large number of nodes, AllReduce requires large number of sequential communication steps, whereas their algorithm requires a single communication step regardless of number of nodes. Their "time per iteration" result support this, at 32 nodes they require less time per iteration than all-reduce. However, I don't understand why time per iteration grows with number of nodes, I expect it to be constant for their algorithm.

The improvements seem to be quite modest which may have to do with AllReduce being very well optimized. In fact, their experimental results speak against using their algorithm in practice -- the relevant Figure is 2a and their algorithm seems to be worse than AllReduce.

Suggestions:
- I didn't see motivation for particular choice of mixing matrix they used -- directed exponential graph. This seems to be more complicated than using fully-connected graph, why is it better?
- From experiment section, it seems that switching to this algorithm is a net loss. Can you provide some analysis when this algorithm is preferrable
- Time per iteration increases with number of nodes? Why? Appendix A.3 suggests that only a 2-nodes are receiving at any step regardless of world-size

---

> ### Author Response · Authors · 2018-11-15
> **Thank you for your feedback (Response Part 1)**
>
> Thank you for your feedback, for finding that our paper has a good balance between theory and practice, and for finding our experiments to be reasonably comprehensive.
>
> *About ‘improvements seem to be quite modest’:*
> We first want to emphasize that the aim of Stochastic Gradient Push (SGP) is to make distributed training less sensitive to low-bandwidth/high-latency links (i.e., to reduce the amount of tight coupling among nodes). Therefore, we expect the benefits of SGP to show in a communication-bound  scenario, which is a fairly common real-world setup as pointed out by AnonReviewer1.  For example, typical Amazon EC2 instances have a communication bandwidth of 5Gbps—25Gbps. (source: https://aws.amazon.com/blogs/aws/the-floodgates-are-open-increased-network-bandwidth-for-ec2-instances/)
>
> We demonstrate the benefit of SGP for a communication bound, low-bandwidth interconnect (Ethernet 10Gbps) in the experiment described section 5.1. In particular in Figure 1a., we show that SGP is more than 3 time faster than SGD when training on 256 GPUs, which we think is a significant speed-up, while the top-1 validation accuracy of SGP remains within 1.2% of AllReduce SGD.
>
> To provide a complete empirical evaluation and also show limitations of SGP, we also investigate a high-bandwidth (InfiniBand) scenario which is not communication bound for the size of model used in our experiments. We agree that the improvements observed are quite modest in this scenario; since communication is not a bottleneck, we did not expect SGP to outperform AllReduce SGD. The goal of this experiment was to illustrate that SGP is not significantly slower than AllReduce SGD in a high-bandwidth scenario (for which AllReduce is heavily optimized). However, given that the resulting accuracy of AllReduce SGD is better than SGP, it is clear that AllReduce SGD should be preferred in such a setting. We will revise the paper to clarify this point further.
>
> *About the 'motivation for particular choice of mixing matrix':*
> As you noted, we chose to use the directed exponential graph in our experiments. Of course, there are lots of possible alternatives including a fully-connected (complete) graph. The choice of mixing matrix (or really, the sequence of mixing matrices obtained by cycling through neighbors) impacts how well-synchronized the values at different nodes remain after approximate distributed averaging. To understand this choice, let's focus on simply averaging using linear iterations x(k+1) = P(k) x(k), ignoring that additional gradient term being injected at each node for optimization.
>
> Since x(k) = P(k-1) P(k-2) ... P(1) P(0) x(0), the worst-case averaging error after k steps is related to the second largest singular value of the product of matrices P(k-1) P(k-2) ... P(1) P(0); see, e.g., the reference Nedic et al. (2018) cited in our paper for a detailed derivation. Note that, since the matrices are column stochastic, the second largest singular value lies in the interval [0,1].
>
> The directed exponential graph has an expander-like quality: from any starting vertex, one can get to any destination after at most log2(n) steps because one can always take a step which reduces the distance to the destination by half. This also translates to good mixing properties. For a 32-node directed exponential graph, each node has 5 neighbors. After every product of 5 consecutive matrices P(4) P(3) P(2) P(1) P(0), where matrix P(k) corresponds to the step where every node i sends a message to node (i + 2**k) mod n, the second largest singular value of the product is at machine precision; i.e., if nodes were only averaging, then they would effectively have the exact average. On the other hand, if we cycle through edges in the complete graph then after 5 steps the second largest singular value is roughly 0.6, meaning that in the worst case, the difference || x(k) - avg(0) || <= 0.6 ||x(0) - avg(0)||, where avg(0) is the average of the initial values at all nodes. Hence, cycling through edges of the fully-connected graph would lead to nodes being much less well-synchronized over shorter times, or taking many more iterations to be well-synchronized. In the context of stochastic optimization, this can be interpreted as having additional noise in the step direction which, we believe, will lead to worse accuracy. This is precisely what we observed in other experiments, not reported in the paper.
>
> In response to your suggestion we will add this motivation for using the directed exponential graph to the paper.

---

> > ### Author Response · Authors · 2018-11-15
> > **Thank you for your feedback (Response Part 2)**
> >
> >
> > *About the 'Time per iteration increases':*
> > We agree that in an ideal case, the time per iteration of SGP should be constant as we increase the number of nodes. We do observe a constant time per iteration for different networks sizes in the high-latency experiment. However, SGP is still a synchronous method, so other factors such as the cluster load could affect the timing. In particular, at each iteration every node waits (blocks) to receive a message from one neighbor. Hence, if one node or communication link is slow, the node waiting for that message will be delayed, but the rest of the network can continue processing. In this way, small delays (e.g., due to reading data from disk, communicating, computing gradients...) get averaged out over time. Those factors could be more prominent in the low-latency scenario, as discussed in section 5.3, where the overall time per iteration is lower.
> >
> > We have run further experiments in the low-latency (InfiniBand) scenario to verify this hypothesis. First we re-ran AllReduce SGD and SGP 5 times, using 16 nodes (128 GPUs), to have a better sense of the timing variability. We observed that the time-per-iteration of SGP was actually slightly faster than SGD on average, and also observed significant time variability between the runs. (SGP: avg. time/iter: 0.279; avg. max time: 0.295; avg. min time: 0.264. AllReduceSGD: avg. time/iter: 0.287; avg. max time: 0.295; min time: 0.281).
> >
> > We found that the timing variability was mainly caused by the data loading. To better isolate the effects of data-loading, we ran experiments on 32, 64, 128 GPUs where we first copied the data locally on every node. In that setting, we observe that the time-per-iteration of SGP remains approximately constant as we increase the number of nodes in the network, while the time for AllReduceSGD increases.
> >
> > We will include these additional results in the revised version of the paper.

---

> ### Author Response · Authors · 2018-12-10
> **Follow-up**
>
> Thank you again for your comments and suggestions. Have our responses and the changes we made to the manuscript addressed all of your concerns?

---

### Author Response · Authors · 2018-11-21
**Revised Version 1.0**

Thanks again for your feedback. We uploaded a new revision of the paper to take into account the reviewers' remarks. In particular:

* AnonReviewer1:
    * We added a comparison with the asynchronous AD-PSGD method on the high-latency interconnect in section 5.1.
        * We find that AD-PSGD exhibits virtually no accuracy degradation relative to D-PSGD over the range from 32—256GPUs, and provides a substantial speedup in the high latency case, bolstering the findings in Lian et al. 2018
        * We find that AD-PSGD runs slightly faster than SGP at the expense of slightly lower training/validation accuracy (except in the 256GPU case where the accuracies are very similar; cf. Tables 1 and 2 in Section 5.1).
        * note: in our original reply we had implemented AD-PSGD exactly as described in Appendix A of Lian et al. 2018. However, we noticed that AD-PSGD was bottlenecked by the Python GIL in this multithreaded case, so we implemented a completely concurrent multi-processed version of AD-PSGD in PyTorch.  The algorithm is implemented as a PyTorch module in a similar manner as torch.distributed.DistributedDataParallel, and, as stated previously, we plan to release our implementations to foster more research in this area. Of course it is difficult to compare the runtime of implementations on different systems, but just for perspective, our new implementation of AD-PSGD runs at a rate of ~226s/epoch and achieves an accuracy of ~76% in the 64GPU case, compared to the rate of 264s/epoch and accuracy of ~74% presented in Lian et al. 2018 for a similar 10Gbps network. This corresponds to a 1 hour reduction in training time with a 2% improvement in the top-1 validation accuracy.
        * Again we emphasize that this asynchronous line of work is orthogonal, and that by combining the two approaches (leveraging the PushSum protocol in an asynchronous manner), one can expect to further speed up SGP. We leave this as a promising line of investigation for future work.
    * We updated the description of AD-PSGD in related work section to better reflect the findings of Lian et al. 2018.
    * We reported the number of iterations K used by the different training runs in appendix B.1
* AnonReviewer2:
    * We clarified the definition of the mixing matrix sequence, P^k, and added a theoretical motivation and succinct empirical support for the use of directed exponential graphs in Appendix A.1, including a comparison with randomly sampled mixing matrix sequences.
* AnonReviewer3:
    * We added a theoretical motivation for using the directed exponential graph for mixing matrices in Appendix A.1, and supplanted the discussion with a succinct empirical comparison with mixing matrices constructed from fully-connected graphs.
    * We updated the low-latency network experiment (section 5.2) to clarify that SGP benefits are more prominent in high-latency/low-bandwidth communication-bound scenarios.
    * We also added a discussion section 5.2 and additional experiment in appendix B.3 showing that when we isolate the effect of data loading, the time-per-iteration of SGP is approximately constant as we increase the network sizes, while the time for AllReduceSGD increases.
    * We added new scaling plots in appendix B.4 comparing SGP and SGD throughput on low-latency and high-latency interconnect. Those plots highlight the robustness of SGP to high-latency interconnect.


In addition to those revisions, we also provided a minor addition to the theory:
In the statement of Theorem 1 in equation (6) we add the constraint K>=n. In the proof of the Theorem 1 in section C.3.1 we restrict the upper bound of parameter \gamma to be \min\left\{\dfrac{(1-q)^2}{60L^2C^2},1 \right\}$, rather than just \dfrac{(1-q)^2}{60L^2C^2}, which we had before. This small modification was made to handle the case of very small parameter C. The rest of the proof is the same as the original version.

---

### Meta-Review · Area_Chair1 · 2018-12-13
**Paper**

**Confidence:** 5
**Recommendation:** Reject

**Metareview:**

The reviewers liked the paper in general but the empirical evaluation lacks studies on a wider range of different data sets.